# An all-to-all approach to the identification of sequence-specific readers for epigenetic DNA modifications on cytosine

Guang Song [1,7], Guohua Wang [2,7], Ximei Luo [2,3,7], Ying Cheng[4,5], Qifeng Song[1], Jun Wan [3], Cedric Moore[1], Hongjun Song [6], Peng Jin [4], Jiang Qian [3✉] & Heng Zhu [1✉]

Epigenetic modifications of DNA play important roles in many biological processes. Identifying readers of these epigenetic marks is a critical step towards understanding the underlying mechanisms. Here, we present an all-to-all approach, dubbed digital affinity profiling via proximity ligation (DAPPL), to simultaneously profile human TF-DNA interactions using mixtures of random DNA libraries carrying different epigenetic modifications (i.e., 5-methylcytosine, 5-hydroxymethylcytosine, 5-formylcytosine, and 5-carboxylcytosine) on CpG dinucleotides. Many proteins that recognize consensus sequences carrying these modifications in symmetric and/or hemi-modified forms are identified. We further demonstrate that the modifications in different sequence contexts could either enhance or suppress TF binding activity. Moreover, many modifications can affect TF binding specificity. Furthermore, symmetric modifications show a stronger effect in either enhancing or suppressing TF-DNA interactions than hemi-modifications. Finally, in vivo evidence suggests that USF1 and USF2 might regulate transcription via hydroxymethylcytosine-binding activity in weak enhancers in human embryonic stem cells.

[1] Department of Pharmacology and Molecular Sciences, Johns Hopkins University School of Medicine, Baltimore, MD, USA. [2] School of Computer Science and Technology, Harbin Institute of Technology, Harbin, Heilongjiang, China. [3] Department of Ophthalmology, Johns Hopkins University School of Medicine, Baltimore, MD, USA. [4] Department of Human Genetics, Emory University School of Medicine, Atlanta, GA, USA. [5] Institute of Biomedical Research, Yunnan University, Kunming, Yunnan, China. [6] Department of Neuroscience and Mahoney Institute for Neurosciences, University of Pennsylvania, Philadelphia, PA, USA. [7] These authors contributed equally: Guang Song, Guohua Wang, Ximei Luo. ✉email: jiang.qian@jhmi.edu; hzhu4@jhmi.edu

In mammals, DNA methylation of cytosine at the 5-position (mC) serves as a major epigenetic mechanism to regulate gene transcription and plays a critical role in many cellular functions, such as genomic imprinting, X-chromosome-inactivation, repressing transposable elements, and regulating transcription[1,2]. Aberrant DNA methylation is a hallmark of many human diseases and cancers. Indeed, abnormal methylation has become a potential biomarker for cancer detection, diagnosis, and prognosis used in liquid biopsy[3]. In addition, dysregulation of DNA methylation has been found to be associated with other diseases, such as schizophrenia and autism spectrum disorders[4,5].

Recent efforts revealed that mC can be oxidized by three mammalian ten eleven translocation (Tet) proteins to form 5-hydroxymethylcytosine (hmC)[6]. Sequential oxidation by the Tet enzymes can further convert hmC to 5-formylcytosine (fC) and 5-carboxycytosine (caC), which eventually leads to active DNA demethylation[7–9]. However, it remains largely unknown whether hmC, fC, and caC merely serve as intermediates in the process of active DNA demethylation or whether they have their own physiological functions[10]. Recently, genome-wide mapping of hmC revealed that hmC is present in many tissues and cell types, and is especially enriched in embryonic stem cells and neurons[11–15]. In addition to hmC, fC was also found as a stable DNA modification in mammalian genomes[8,16–18]. These studies indicated that the oxidized forms of mC might have their own physiological functions, and identification of their "readers" and "effectors" might help elucidate their roles in various biological processes[19–23].

Methylated cytosine can also remain asymmetric in mammalian genomes, and such sites are referred to as hemi-methylated[24]. A prevailing view is that hemi-methylation is transient and occurs, perhaps, by chance, and that the fate of hemi-methylated DNA is to become fully methylated or unmethylated by replication-coupled dilution[25]. However, two recent studies revealed that ~10% of CpGs remain stably hemi-methylated in embryonic and trophoblast stem cells[26,27]. In a recent study, Xu and Corces demonstrated that elimination of hemi-methylation caused a reduction in the frequency of CTCF/cohesion interactions at these loci, suggesting hemi-methylation as a stable epigenetic mark regulating CTCF-mediated chromatin interactions[24]. Intriguingly, they reported that hemi-methylated sites could be inherited over several cell divisions, suggesting that this DNA modification could happen by design and be maintained as a stable epigenetic state. Furthermore, in a non-CpG context (i.e., CpA, CpT, and CpC) mC is asymmetrical and hemi-hmC can arise via Tet oxidation[11,28–30]. As we and others have shown, mCpA is found more often in gene bodies and represses gene transcription through MeCP2 binding[31–33]. In the context of the palindromic CpG dinucleotide, hmCpG presumably exists in a fully hydroxymethylated form in cells; however, they can transiently become hemi-hmC after semi-conservative DNA replication[34]. Following this logic, hemi-fC and hemi-caC should also exist, at least transiently, in mammalian genomes.

Despite acceleration of efforts to map mC and hmC at single base pair resolution in various biological processes and species, it remains a challenge to establish the causality between different types of epigenetic DNA modifications and physiological outcomes. Therefore, the identification of 'readers' and 'effectors' for mC, hmC, fC, and caC in both symmetric and hemi-forms will serve as a critical stepping stone to translate epigenetic signals into biological actions and to decipher the epigenetic 'codes' governing biological processes.

In the past, various types of high-throughput technology, such as protein binding microarrays (PBM), transcription factor (TF) arrays, yeast one-hybrid, SMiLE-seq and high-throughput SELEX, were developed to profile protein-DNA interactions (PDIs) mostly in the absence of any epigenetic modifications[35–39]. To identify potential readers for mC, our team was among the first to survey the majority of human TFs with 154 symmetrically methylated DNA probes[38]. Later, Taipale and colleagues individually screened several hundred TF proteins against symmetrically methylated random DNA libraries using SELEX[38,40,41]. In addition, generic DNA sequences, carrying symmetrically modified mC, hmC, fC, or caC, were used to pull down and identify potential binding proteins through MS/MS analysis[20,42].

Although these high-throughput efforts have generated massive amounts of data, none of them can be multiplexed to exhaustively survey both the protein (e.g., the entire human TF family) and DNA spaces (e.g., a random DNA library) simultaneously. Due to these design limitations, no systematic efforts have been reported to identify readers for symmetric-hmC, -fC, or -caC modifications, let alone readers for hemi-mC, -fC, or -caC modifications.

To overcome this huge technology bottleneck, herein we report the invention and application of digital affinity profiling via proximity ligation (i.e., DAPPL) as an all-to-all approach to exhaustively survey human TFs with mixtures of random DNA libraries carrying mC, hmC, fC, or caC modifications in either symmetric- or hemi-form to identify sequence- and modification-specific readers. Using specific DNA fragments to covalently barcode each of the 1239 unique human TFs and co-factors, we could connect the identity of a protein to its captured DNA fragments via proximity ligation in a highly multiplexed reaction (e.g., 192 proteins vs. a mixture of five random DNA libraries). Using this DAPPL approach, we identified numerous readers for symmetric-hmC, -fC, and -caC, and hemi-mC, -hmC, -fC, and -caC. We observed that all four modifications could either enhance or suppress TF-DNA interactions and, in some cases, alter TF-binding specificity. We also observed and experimentally validated that symmetric modifications have a stronger effect in either enhancing or suppressing TF-DNA interactions than hemi-modifications. Finally, in vivo evidence suggested that USF1 and USF2 might regulate transcription via hydroxymethylcytosine-binding activity in weak enhancers in human embryonic stem cells.

## Results

**Establishment of digital affinity profiling via proximity ligation.** To create an all-to-all approach for unbiased profiling of TF-DNA interactions, we invented the DAPPL approach, which was achieved in five major steps (Fig. 1). First, human proteins were purified as GST fusions and kept on glutathione beads, while a set of DNA barcode sequences were designed such that any two single nucleotide mis-incorporations, insertions, and/or deletions would not lead to misassignment of protein identity (Supplementary Fig. 1a-b). Second, each protein was covalently tethered to a single-stranded DNA (ssDNA) oligo (i.e., Anchor Oligo) through a thiol–maleimide "click" chemistry linkage; the Anchor Oligos of each protein were then annealed to the 3′-end of a unique barcode oligo, and converted to double-stranded DNA (dsDNA) with a DNA polymerization reaction (using DNA polymerase I Klenow fragment) on beads (Supplementary Fig. 1c). After removal of the free DNA oligos, an aliquot of beads for each barcoded protein was mixed together to generate a protein mixture. Third, a random N-mer dsDNA library was synthesized and capped by a BsaI restriction site to allow for proximity ligation and a fixed sequence (i.e., Primer 2) for PCR priming (Fig. 1). Fourth, to carry out the DAPPL reactions, the N-mer DNA library was incubated with the barcoded protein-bead mixture, followed by stringent washing steps to remove unbound DNA. After a crosslinking step, Golden Gate Assembly

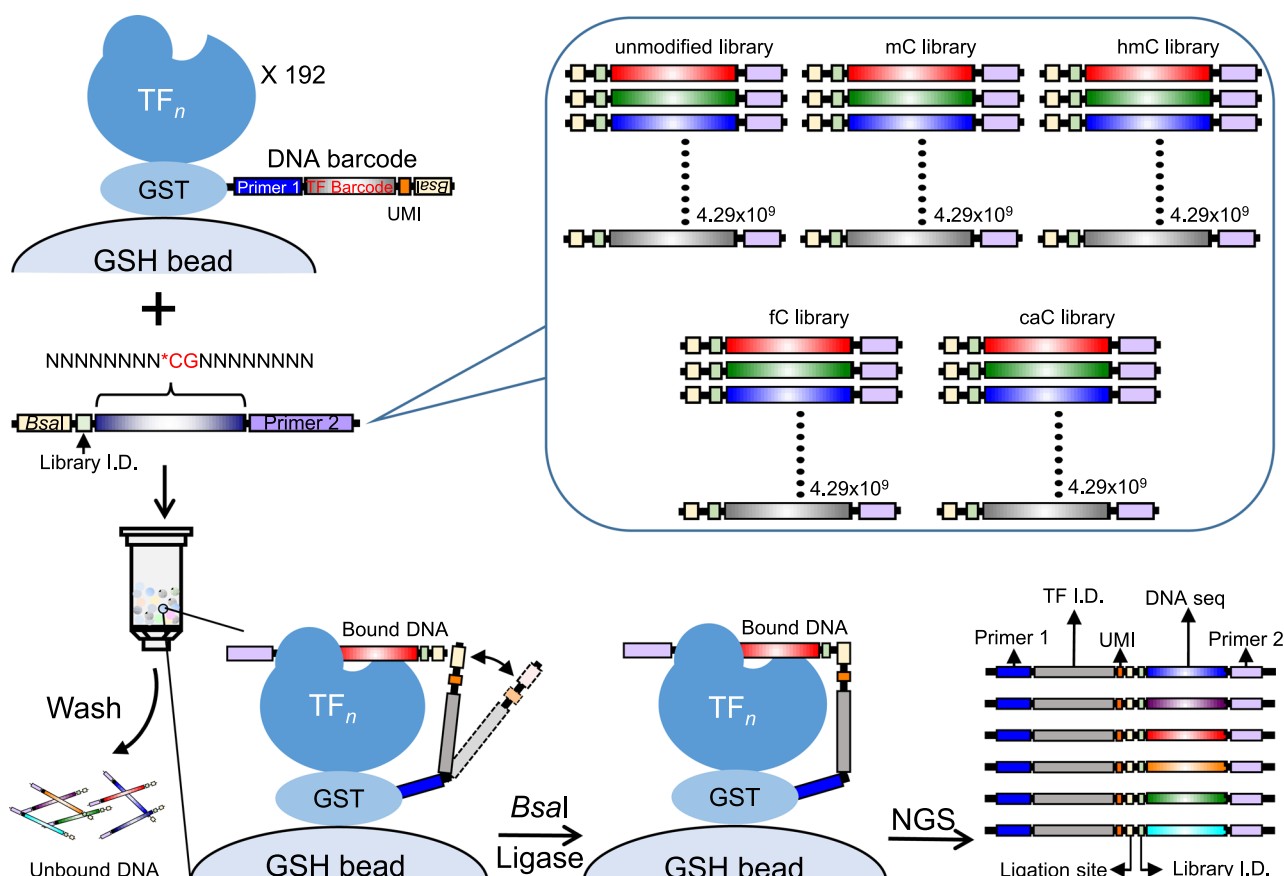

**Fig. 1 Schematics of identifying epigenetic modification readers by DAPPL.** Principle and execution of the digital affinity profiling via proximity ligation (DAPPL) approach. The principle of the DAPPL approach is to utilize the unique DNA barcodes tethered to TF proteins as identifiers to de-convolute the DNA sequences that a given TF captures in a highly multiplexed binding assay. First, each purified TF protein on glutathione beads was individually conjugated with a unique DNA barcode. Second, barcoded TF proteins on beads are mixed and incubated with a mixture of five randomized DNA libraries (the multicolored middle bricks of each DNA means various randomly synthesized DNA sequence species) carrying either symmetric or hemi-modifications. Third, after removal of unbound DNA fragments, a TF-capture DNA fragment can be ligated to the DNA barcode conjugated on that particular TF via Golden Gate Assembly due to the close proximity of the TF barcode DNA. Fourth, the ligated products can be PCR-amplified by utilizing the constant sequences attached to the TF barcodes and one end of the DNA library. Finally, the sequences obtained with Next-Generation sequencing will be de-convoluted and analyzed with our bioinformatics tools (see Supplementary Fig. 3 for more details).

reactions[43] were performed to ligate the DNA barcodes of the proteins to their captured DNA fragments. Finally, the ligated DNA products were PCR-amplified with specific primers and a sequencing library was constructed for next-generation sequencing (Fig. 1).

To optimize the DAPPL approach, we decided to focus on the human ETS TF subfamily because almost all of them have well-characterized binding consensus sequences that can be used to benchmark the DAPPL approach (Supplementary Fig. 1d). First, we purified a total of 31 ETS proteins (28 unique) and successfully DNA-barcoded them as examined with both Coomassie and ethidium bromide staining (Supplementary Fig. 1e; Supplementary Data 1). We included CRX and HSF1 as additional positive controls, as well as BCAT1, COPE, and GST as negative controls. After generating a bead mixture of all of the barcoded proteins, we incubated the pooled beads with a random 16-mer dsDNA library in triplicate. The ligated DAPPL products were PCR-amplified and subjected to Next-Generation sequencing (Supplementary Fig. 1d).

A total of ~33.8 million reads were obtained, 50.0% of which precisely contained all the elements of a DAPPL product (Supplementary Fig. 2a). Any errors in the TF barcode sequences, Primer 1, Primer 2, ligation site, library ID, or the length of UMI would disqualify a DAPPL product (Supplementary Fig. 2b). The

median number of qualified unique reads per protein was 60,220 (Fig. 2a). As expected, the three negative control proteins, namely COPE, BCAT1, and GST, consistently captured the lowest numbers of reads across all three replicates (Fig. 2a). To determine the binding consensus for a given TF, we evaluated its 6-mer frequencies by sliding a 6-mer window along the 16-mer sequences it captured (Supplementary Fig. 3). To remove possible noise signals resulting from nonspecific binding to either the glutathione beads or the GST tag, we compared the 6-mer subsequence frequencies of each TF with those obtained with the barcoded GST included as a negative control. Those 6-mer subsequences that were found preferentially enriched by a given TF were extracted and mapped back to the original 16-mer sequences, which were then used to deduce candidate consensus sequences using HOMER motif analysis[44]. Only those motifs with a false discovery rate of 0.05 were considered significant (Supplementary Fig. 3).

To benchmark the DAPPL approach, we first examined its reproducibility by comparing the frequencies of 6-mers bound by the same proteins between the triplicates. We found that the vast majority of them showed high reproducibility. An example, ELF2, is shown in Fig. 2b. Next, we calculated the Pearson correlation coefficients (PCC) of the 6-mer frequencies for each protein between replicates 1 versus 2, 1 versus 3, and 2 versus 3. All of

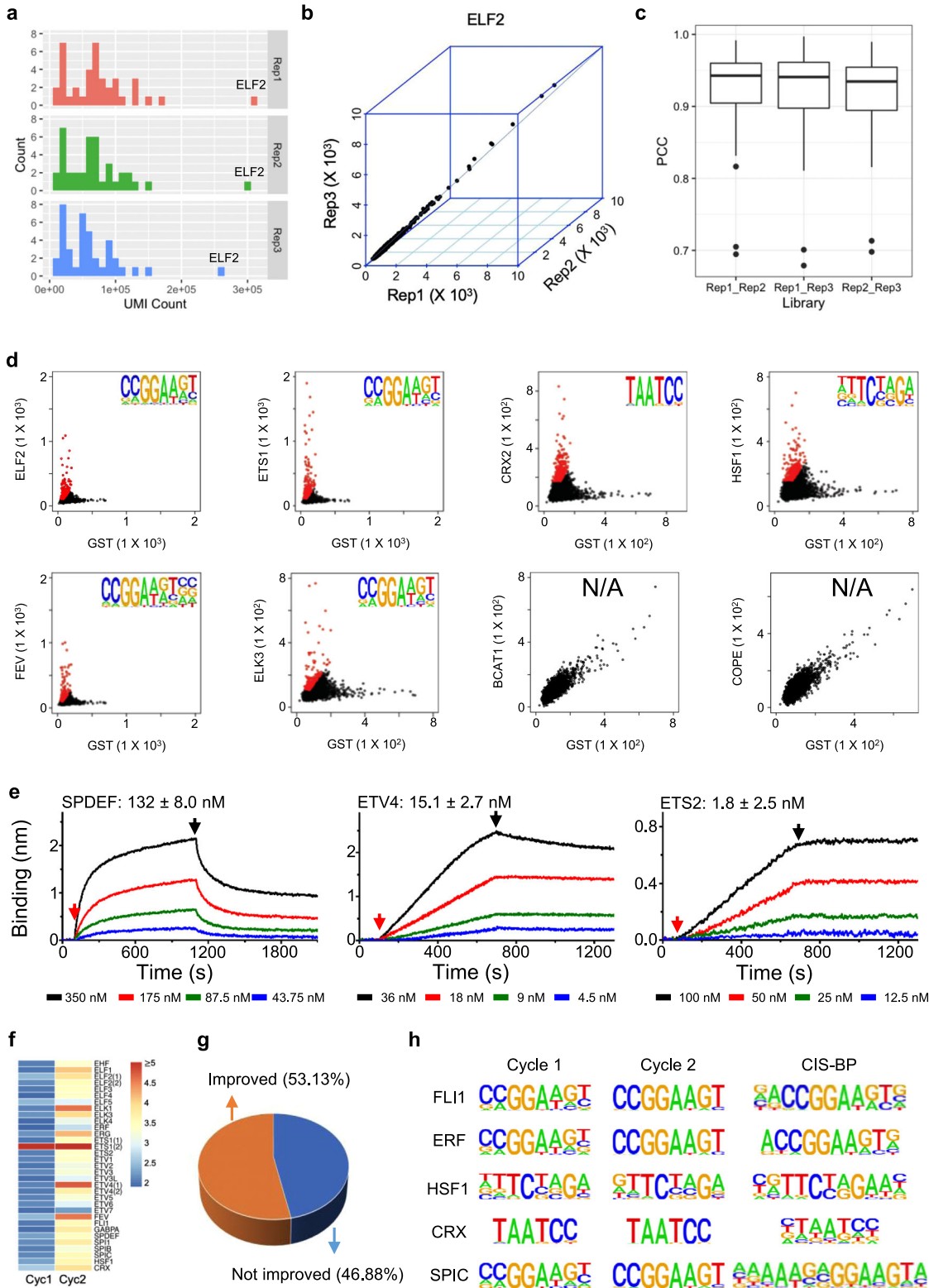

them showed high PPC values, ranging from 0.701 to 0.997 (Fig. 2c).

To extract the 6-mers that were enriched for each protein, we compared the 6-mer frequencies between a given TF and the GST negative control. For the ETS members and two positive controls of different TF subfamilies, a large number of 6-mers were highly enriched (marked in red, Fig. 2d). In contrast, the two negative controls (COPE and BCAT1) failed to produce any enriched

6-mers, suggesting that the captured DNA sequences were likely due to nonspecific interactions. The enriched 6-mers were used to identify the binding consensus and all of the tested ETS members, as well as CRX and HSF1, produced significant consensus sequences. A heatmap was also generated to help visualize the 6-mers enrichment for each protein tested (Supplementary Fig. 4). To assess the quality of the obtained consensus sequences, we employed Tomtom to compare the similarity of the obtained

**Fig. 2 Development of the DAPPL approach using ETS members as a benchmark. a** Distribution of unique molecular identifier (UMI) reads per protein obtained from the three replicated DAPPL (digital affinity profiling via proximity ligation) reactions. The three negative control proteins, COPE, BCAT1, and GST, consistently showed the lowest read numbers in all three replicates, while ELF2 showed the highest read number across. **b** Comparison of the 6-mer frequencies of ELF2 in three independent DAPPL reactions. **c** Boxplot analysis of pairwise Pearson correlation coefficients among the triplicate. The minima values of comparison of these three groups are 0.6947, 0.6789, and 0.698, respectively. The maxima values are 0.9918, 0.9971, and 0.9897. The median values are 0.9427, 0.9408, and 0.9346. The 3rd quantiles are 0.96, 0.9612, and 0.9544. The 1st quantiles are 0.9046, 0.8975, and 0.8945. The down whisker values are 0.8314, 0.8107, and 0.8157. **d** Examples of scatterplot analysis for ETS members, the two positive and two negative control proteins. The red dots are the enriched 6-mers used to identify the consensus sequences for these proteins. The recovered consensus sequences are also shown. **e** Examples of binding kinetics of three ETS members to the same consensus sequence obtained with the OCTET system. X-axis represents time of the kinetics studies, and Y-axis the nanometer shift used to define the biosensor surface changes. The $K_D$ values shown in the figure were the average ± standard error of the obtained $K_D$ obtained at four different TF concentrations shown in different colors. Source data are provided as a Source Data file. **f** Comparison of the enriched 6-mers in cycles 2 and 1. The color represents the average frequency of the enriched 6-mers associated with a particular protein normalized by those obtained with GST. **g** Breakdown of the TFs on the basis of consensus similarity changes in the cycle 2 of the DAPPL reactions. **h** A few examples of consensus sequences are shown. Source data are provided as a Source Data file.

consensus sequences with those archived in CIS-BP. The resultant $P$ values ranged from $3.7 \times 10^{-9}$ to $3.1 \times 10^{-3}$, indicating that DAPPL reactions could faithfully recover the known motifs with high success rate and high quality (Supplementary Data 2, 3).

To address the concern of potential competition during the DAPPL reactions, we randomly selected 14 ETS members to examine their binding kinetics to the same consensus sequence (i.e., 5′-ACTTCCGG) using a label-free, real-time method (i.e., OCTET). The observed $K_{on}$ and $K_{off}$ ranged from $5.2 \times 10^3 \, M^{-1}s^{-1}$ to $2.1 \times 10^5 \, M^{-1}s^{-1}$, and $2.0 \times 10^{-5} \, s^{-1}$ to $2.1 \times 10^{-3} \, s^{-1}$, respectively, and the deduced $K_D$ ranged from 1.8 to 132 nM (Fig. 2e and Supplementary Fig. 5a–n). Importantly, no significant correlation was observed between the read number and the obtained affinity values (Supplementary Fig. 5o). For example, among the 14 ETS members with measured affinity values, ETS2 showed the strongest affinity (1.8 nM) that was 73-fold stronger than SPDEF (132 nM), while the average read numbers of the two was only 15-fold (=509,195/33,322) different. This analysis indicated that DAPPL reactions were sensitive enough to detect binding events for TFs with relatively lower affinity.

To evaluate whether multiple rounds of selection could further improve the performance of the DAPPL reactions, we implemented a two-cycle selection and performed the DAPPL reactions at the end of the second cycle, a strategy similar to SELEX[38,40,41]. After next-generation sequencing, the same approach was applied to identify consensus sequences for all the proteins. We compared the frequency of the binding 6-mers (i.e., red dots in Fig. 2d) between cycle 1 and cycle 2 and found that these 6-mers were indeed enriched in cycle 2 for every ETS protein, as well as the two positive control TFs (Fig. 2f and Supplementary Fig. 6). Using the same approach described above, we were able to determine consensus sequences for all 28 ETS TFs and the two positive controls with the enriched 6-mers obtained in cycle 2. To compare the quality of the consensus sequences of cycle 1 versus cycle 2, we calculated the similarity between these consensus sequences and the known sequences in CIS-BP. Judged by the similarity scores, those consensus sequences obtained in cycle 2 only showed marginal improvement (~53%), suggesting that although the binding sequences were enriched in cycle 2, the DAPPL approach was sensitive enough to generate significant and reliable consensus sequences in one cycle of screening (Fig. 2g). Examples of consensus sequences with similarity scores from high to low are illustrated in Fig. 2h. One plausible explanation is that the ligation step in the DAPPL served as another layer of selection, resulting in further enrichment of the high affinity binding sequences.

Taken together, we developed an all-to-all approach (i.e., DAPPL) that allowed us to identify protein-DNA interactions

in vitro in a highly multiplexed fashion. The quantitative evaluation of DAPPL consensus sequences against those already known has successfully benchmarked the DAPPL approach. Our experiments have suggested that a single round of selection is sufficient to produce high quality consensus sequences, laying the foundation for identifying readers for various epigenetic modifications on cytosine.

**Epigenetic modification-associated TF-binding activities**. To harness the power of DAPPL, we applied it to identify TF-DNA interactions for 1543 human TFs (1239 of which are unique) and co-factors (902 TFs and 337 co-factors) (Supplementary Fig. 1f and Supplementary Data 1) in the context of four epigenetic modifications, including mC, hmC, fC, and caC in both symmetric and hemi-forms. Because complete modifications of hmC, fC, or caC cannot be obtained with an enzymatic reaction, four random DNA oligo libraries were first synthesized, each of which carried a CpG site with methylation, hydroxymethylation, formylation, or carboxylation flanked by 8-mer random sequences (Supplementary Table 1). Next, they were converted to double-stranded, symmetric and hemi-libraries via Klenow reactions in the presence or absence of the corresponding modified dCTPs, respectively. An equal amount of the four symmetric and hemi-modified libraries (i.e., mC, hmC, fC, and caC) was mixed separately to generate two mixtures of the symmetric and hemi-modified libraries, respectively. An unmodified library of the same design was also synthesized and added to the two library mixtures in an equimolar ratio (Fig. 1).

The two library mixtures carrying the symmetric or hemi-modifications were then separately incubated with the TF mixtures to carry out the DAPPL reactions, as described above. Using the same computational approach, 44, 97, 84, and 107 TFs were identified that recognized specific DNA consensus sequences containing symmetric modifications of mC, hmC, fC, and caC, respectively (Fig. 3a and Supplementary Data 4, 5). Similarly, 99, 103, 118, and 139 TFs were identified to recognize specific DNA consensus sequences containing hemi-modifications of mC, hmC, fC, and caC, respectively (Fig. 3b and Supplementary Data 4, 5). The corresponding heatmaps can be found in Supplementary Data 6.

Venn diagram analysis revealed that many TFs could recognize both symmetric and hemi-modifications (Fig. 3c). Interestingly, the identified binding activities to both symmetric- and hemi-mC, -hmC, -fC, and -caC modifications are observed across all the major TF subfamilies (Fig. 3d and Supplementary Data 4, 5). Overall, more TFs were found to interact with hemi-modified DNA consensus sequences than symmetrically modified sequences. Furthermore, no

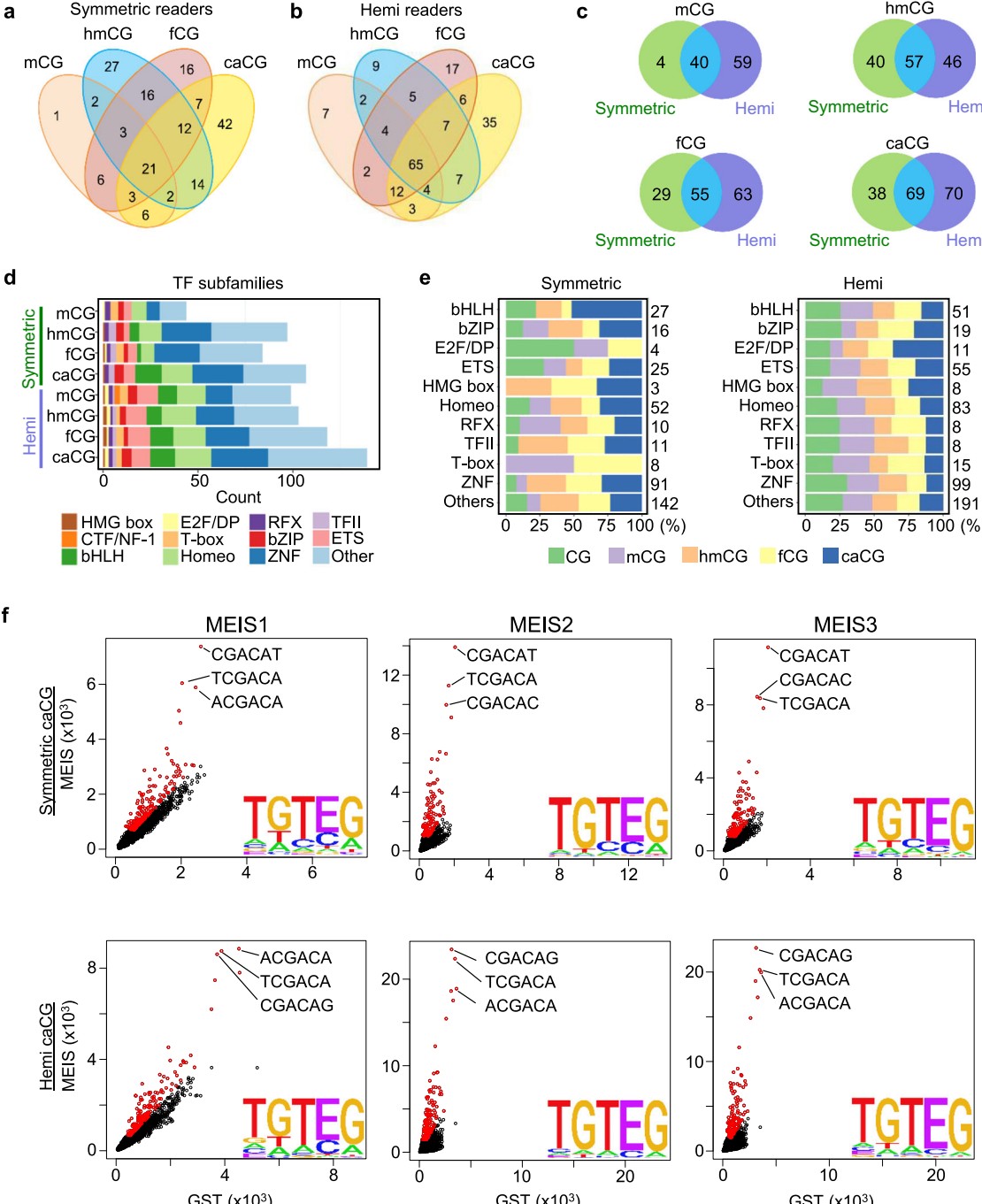

**Fig. 3 Identification of readers for various epigenetic modifications. a** Venn diagram analysis of identified readers for mC, hmC, fC, and caC modifications in symmetric form. **b** Venn diagram analysis of identified readers for mC, hmC, fC, and caC modifications in hemi- form. **c** Venn diagram analysis between symmetric and hemi-modification readers in the context of mCpG, hmCpG, fCpG, and caCpG modifications. **d** The identified binding activities to both symmetric and hemi-mC, -hmC, -fC, and -caC modifications are observed for all the major TF subfamilies shown in different colors. More TFs were found to interact with hemi-modified DNA consensus sequences than symmetrically modified sequences. **e** No significant preference to a particular symmetric or hemi-modification was observed within each major TF subfamily shown in different colors. **f** Highly conserved homologs often recognized highly similar consensus sequences and shared the preference for the same modifications. For example, MEIS1/2/3 all recognized a similar consensus carrying caC in a sequence of 5′-TGTcaCG in both symmetric and hemi-forms. The scatterplots display the frequencies of 6-mer subsequences obtained with the MEIS1/2/3 (*y*-axis) and GST tag control (*x*-axis). Letter E in the consensus sequences represents caC. The top three 6-mer sequences are highlighted. Source data are provided as a Source Data file.

significant preference for a particular symmetric or hemi-modification type was observed within each major TF subfamily (Fig. 3e).

Perhaps not surprisingly, we observed that highly conserved homologs often recognized highly similar consensus sequences and shared the same preference for a particular modification, indicating that these discovered activities are conserved among closely related paralogs. For example, MEIS1/2/3 all recognized a similar consensus carrying caC in a sequence of 5′-TGTcaCG in both symmetric and hemi-forms (Fig. 3f).

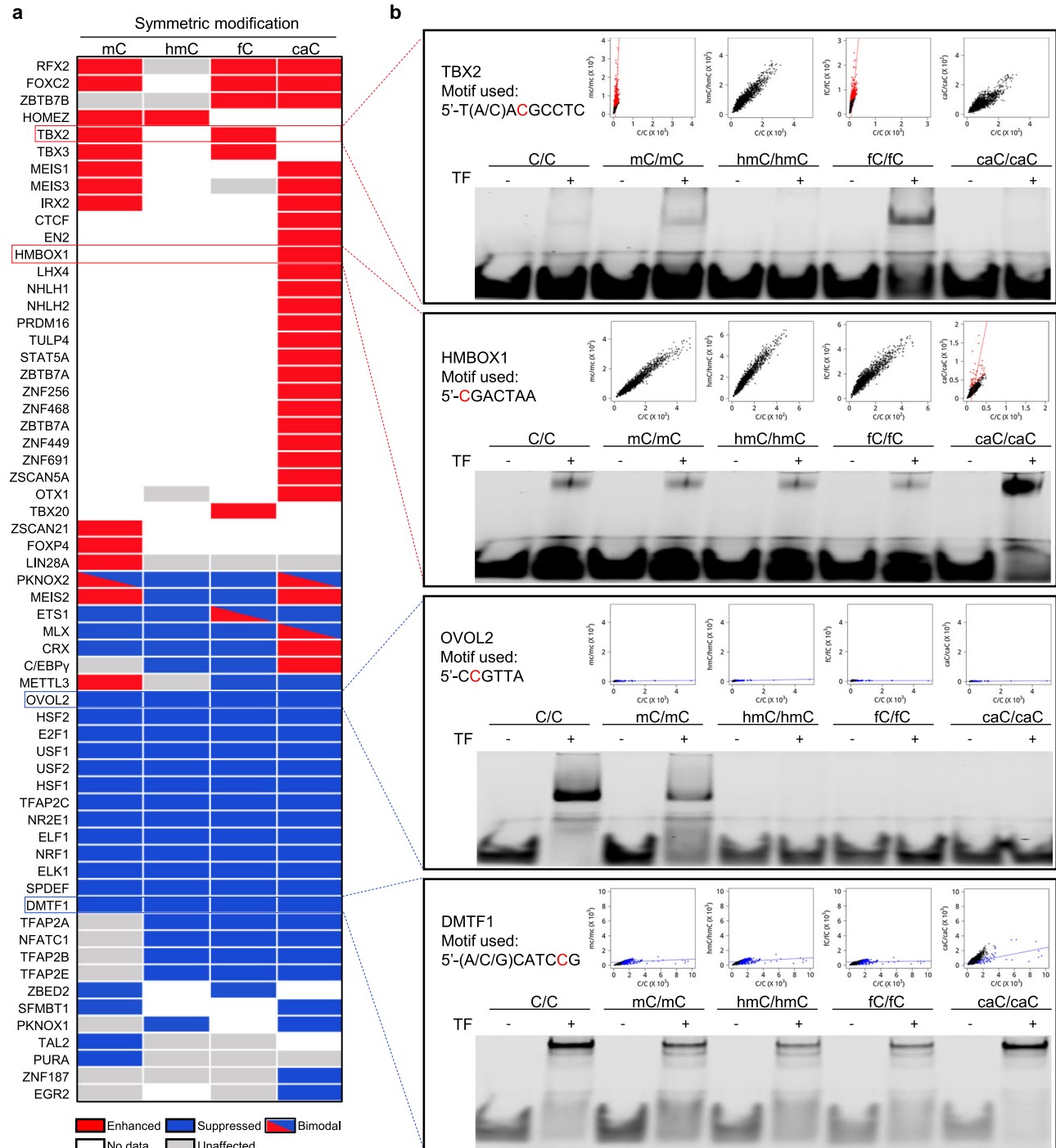

**Fig. 4 Three scenarios of symmetric modification impact on TF-DNA interactions. a** A given TF-DNA interaction can be enhanced (red bricks), suppressed (blue bricks), or unaffected (gray bricks) by any of the four symmetric epigenetic modifications. Note that some modifications could enhance and suppress TF-DNA interactions depending on the sequence context (bimodal). The same scenarios were also observed for hemi-modifications (Supplementary Data 7, 8, and Supplementary Fig. 7). **b** Validation of modification-enhanced (i.e., TBX2 and HMBOX1) or suppressed (i.e., OVOVL2 and DMTF) TF-DNA interactions using electrophoretic mobility shift assays (EMSA). The sequences of the DNA probes used in the EMSA are shown in each box with the modified C labeled in red. The protein concentrations of TBX2, HMBOX1, OVOL2, and DMTF1 were 26.2, 355.6, 563.3, and 156 nM, respectively.

**Impacts of epigenetic modification on TF-DNA interactions.** To examine how different epigenetic modifications could affect TF-DNA interactions, we compared 6-mer subsequence frequencies obtained with each modified library against the unmodified counterparts. Three major trends were observed across all four epigenetic modifications as illustrated in Fig. 4a. First, DNA-binding activity was enhanced for 50 TFs by a particular modification. For example, many 6-mer subsequences captured by HMBOX1 were significantly enriched in the symmetric carboxylated library, suggesting that carboxylation enhanced HMBOX1-DNA interactions. Second, DNA binding was suppressed for 95 TFs by a particular modification. For instance, the

6-mer frequencies of OVOL2 were found to be significantly lower with all four symmetric modification libraries, suggesting that these modifications significantly suppressed OVOL2-DNA interactions. In the third scenario, 25 interactions between TFs and DNA were not significantly affected by any modifications, showing similar binding strength regardless of the modifications. Moreover, some modifications could enhance and suppress the binding activity of a particular TF dependent on the sequence context (labeled as bimodal in Fig. 4a). For example, formylation could enhance the binding strength of ETS1 in a consensus of 5′-CGGAfCGTA, while reducing in a different consensus of 5′-CCGGAAGT (Supplementary Data 7).

Interestingly, different modifications exhibited different impacts on the same proteins. For example, MEIS2 was found to prefer binding methylated and carboxylated DNA, while hydroxymethylation and formylation reduced its binding activity (Row 32; Fig. 4a). Crx, on the other hand, showed strong binding activity to carboxylated DNA while the other three modifications greatly suppressed its binding activity (Row 35; Fig. 4a). Indeed, this phenomenon was observed among many TFs as summarized in Fig. 4a. Overall, 14, 1, 7, and 28 TFs were found to prefer mC, hmC, fC, and caC modifications, respectively; whereas methylation, hydroxymethylation, formylation, and carboxylation suppressed binding activities of 21, 24, 25, and 25 TFs, respectively. On the other hand, 9, 8, 6, and 2 TFs were insensitive respectively to methylation, hydroxymethylation, formylation, or carboxylation (i.e., "No preference" in Fig. 4a). The same major trends were also observed for hemi-modifications (Supplementary Data 8 and Supplementary Fig. 7).

To experimentally validate the observed preferences, purified TBX2, HMBOX1, OVOL2, and DMTF1 were subjected to EMSA analysis because each of them showed a distinct behavior to various epigenetic modifications. Using the consensus sequences obtained from our bioinformatics analysis, TBX2 demonstrated stronger binding strength to methylated and formylated consensus sequences than to the unmodified counterpart, in agreement with the DAPPL results. However, neither hydroxymethylation nor carboxylation enhanced TBX2's binding to the same consensus sequence (Fig. 4b). In the case of HMBOX1, carboxylation in a consensus of 5′-CGACTAA substantially enhanced its binding activity as compared with the unmodified, methylated, hydroxymethylated, and formylated counterparts (Fig. 4b). On the other hand, OVOL2 preferred the unmodified DNA consensus, whereas all the other modifications reduced or completely inhibited its binding activity (Fig. 4b). Similarly, all four modifications were confirmed to suppress DMTF1's binding activity as compared with the unmodified counterpart, although carboxylation showed the least effect (Fig. 4b). Overall, all of the EMSA assays validated our DAPPL results, suggesting that different modifications could impose differential impacts to binding activities of the same TFs.

**Impacts of epigenetic modification on TF-binding specificity**. We also observed that the carboxylated consensus (5′-caCGAC TAA) identified for HMBOX1 was significantly different from its known consensus (5′-TAACTA)[38,45], suggesting that carboxylation could alter TF-binding specificity. This phenomenon is observed across all four modifications. For example, IRX2 preferred 5′-mCGTTA, which is substantially different from its known unmodified consensus 5′-TTACACG (Fig. 5a). Similarly, FOXC2 and HOMEZ showed altered consensus sequences in the presence of different modifications (Fig. 5a). On the other hand, while the binding strength of TBX2 and RFX2 was enhanced by methylation and carboxylation, respectively, neither altered their binding specificity (Fig. 5b).

To systematically examine the impact of the four modifications on TF-binding specificity, we determined the TF consensus sequences that were found preferentially bound to modified sequences in both symmetric and hemi-forms. Specifically, we extracted those 6-mers enriched in modified libraries as compared to the unmodified libraries and constructed the modification-preferred consensus sequences (Supplementary Data 7, 8). We next compared sequence similarity between these obtained consensus sequences with those obtained with unmodified libraries, as well as those archived in CIS-BP[46]. For example, 13 symmetric-mC-preferred consensus sequences were identified, four (30.8%) of which are significantly different from the unmodified counterparts. Similarly, 26.3% of the caC-preferred consensus sequences are significantly different (Fig. 5c). Overall, 27.5% of the modification-preferred consensus sequences are significantly different from the unmodified counterparts.

A similar observation was made for hemi-modification-preferred consensus sequences. For example, hemi-modifications altered binding specificity of NR2E1, YBX1, HMBOX1, and JDP2 (Supplementary Fig. 8a). On the other hand, the binding specificity of PKNOX2 and EGR2 was not affected by hydroxymethylation (Supplementary Fig. 8b). Overall, 35.85% consensus sequences with the four modifications were significantly different from the binding unmodified consensus sequence (Supplementary Fig. 8c).

**Stronger effects imposed by symmetric modifications**. One interesting phenomenon observed was that the extent of modification-enhanced or -suppressed binding activity was greater for symmetric than hemi-modifications in general. For instance, although methylation in both symmetric and hemi-forms enhanced HOMEZ's binding activity, the enriched 6-mers showed steeper slope for symmetric methylation (red lines; Fig. 6a), suggesting that symmetric methylation strengthened HOMEZ-DNA interaction more than hemi-methylation. In the modification-suppressed cases, symmetric modifications also showed a stronger effect. For example, the slope of methylation-suppressed 6-mers of OVOL2 was flatter for symmetric methylation, suggesting that symmetric mCpG nearly abolishes the interactions while hemi-methylation was partially tolerated (Fig. 6b).

To systematically and quantitatively assess this phenomenon, we identified 30 TF-DNA interactions that were enhanced by both symmetric and hemi-modifications and 60 interactions suppressed by both symmetric and hemi-modifications (Fig. 4, Supplementary Fig. 7 and Supplementary Data 7, 8). Next, we quantified the TF-binding preference for a given modification by deducing the slope in the corresponding scatterplot as illustrated in Fig. 6a. Note that a larger slope corresponds with a stronger the preference for that modification. Overall, in 16 (53.33%) of the 30 modification-enhanced cases, symmetric modifications showed higher slopes than the hemi-modifications (Fig. 6a). Similarly, in 58 (96.7%) of the 60 modification-suppressed cases, symmetric modifications showed lower slopes than the hemi-modifications in all four forms (Fig. 6b). These results suggest that symmetric modifications can strongly enhance or suppress TF-DNA interactions while the effects of hemi-modifications are more modest.

To quantitatively confirm the above observation, we selected HOMEZ, TBX2, TBX3, TBX20, and STAT5A, as modification-enhanced cases, to determine their binding kinetics and affinity to DNA fragments with symmetric or hemi-modifications. Taking HOMEZ as an example, we synthesized three versions of DNA oligos carrying unmodified C, mC, or hmC, and then converted to the symmetric and hemi-modified dsDNA probes in the

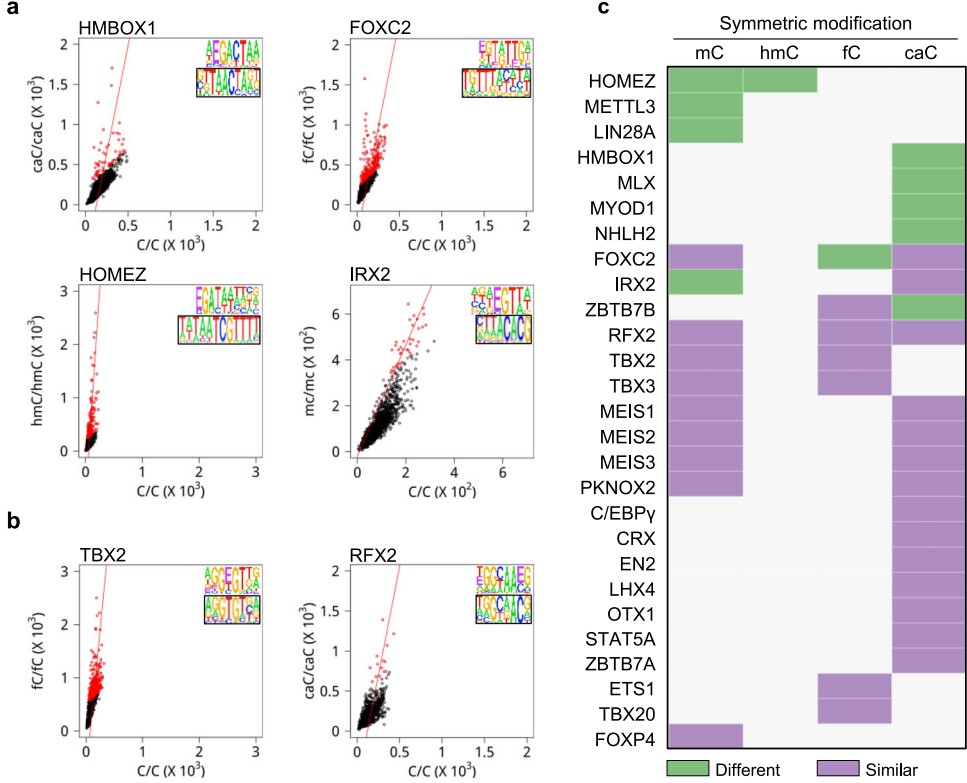

**Fig. 5 Impact of symmetric epigenetic modifications on TF-binding specificity. a** Examples of deviated consensus sequences across all four epigenetic modifications. Known consensus sequences (boxed) in the absence of any modifications are compared with those carrying a modified cytosine. Letter E in the consensus sequences represents modified cytosine. **b** Examples (i.e., TBX2 and RFX2) of TF-binding specificity not affected by cytosine modifications, although with enhanced binding affinity. **c** Summary of impact of different modifications on TF-binding specificity. Green or purple bricks represent the consensus sequences generated from modified DNA libraries are different or similar with that from unmodified DNA libraries, respectively. Source data are provided as a Source Data file.

sequence context of 5′-TATCGATA. Note that the "pure" symmetric sequences were generated by selecting probes such that the only modified C to be incorporated is at the desired modification site. Using the EMSA assays as a semi-quantitative measurement, the symmetric modifications appeared to have a stronger effect than the hemi counterparts (Fig. 6c). Next, we quantitatively determined the binding kinetics and affinity using OCTET instrumentation. After the $K_{on}$ and $K_{off}$ values were determined at different protein concentrations, the $K_D$ values for the DNA probes carrying symmetric-mC, hemi-mC, symmetric-hmC, and hemi-hmC were deduced to be 243, 553, 135, and 345 nM, respectively, while the $K_D$ value with the unmodified probe was 5.90 μM. Similarly, STAT5A, TBX2, TBX3, and TBX20 all showed stronger affinity to DNA probes carrying symmetric-caC and -fC than those carrying the respective hemi-modifications (Fig. 6c, and Supplementary Fig. 9). As an example of a modification-suppressed case, binding and affinity studies also confirmed that symmetrically methylated consensus sequences suppressed the binding activities of OVOL2 more than the hemi-modified sequences (Fig. 6c). Taken together, our binding kinetics and affinity studies confirmed that symmetric epigenetic modifications had a stronger impact in either enhancing or suppressing TF-DNA interactions than hemi-modifications.

**Potential function of hmC readers in human stem cells**. To explore potential physiological functions of identified epigenetic modification readers, we decided to focus on hmC readers because it is more prevalently found in tissues—in particular, human embryonic stem cells—among the oxidized forms of mC. Since the existing hmC map of H1 is of low coverage, we first

employed a selective chemical labeling method[47] to extensively map hmC peaks in human embryonic stem cell H1. A total of 3892 peaks containing hmC were identified, 1783 of which are located in open chromatin regions. Please note that these hmC peaks might contain partially modified sites due to the heterogeneity of the cell population.

Of the 143 unique hmC readers (symmetric and/or hemi) identified in this study, seven have available ChIP-seq data in H1 cells[48]. Therefore, we superimposed the available ChIP-seq peaks with the hmC peaks to identify peaks of overlap (Fig. 7a). We found that the numbers of overlapping peaks were 229, 68, 36, and 46 for USF1, USF2, TAF7, and ATF2, respectively (Fig. 7b). The observed overlapping is significant as determined with a random permutation of ChIP-seq peaks (Fig. 7b; see Methods and Materials for more details). Although peaks of overlap were also observed for NRF1, RXRA, and SRF, none of them were significant using the same criteria. Moreover, using the overlapping peaks, we recovered consensus sequences for both USF1 and USF2 that were significantly similar to those obtained from DAPPL assays (Fig. 7c), suggesting that the two TFs recognize hmC in a similar sequence context in H1 cells.

To examine possible functions of hmC-dependent TF-DNA interactions, we investigated the chromatin status of the overlapping peaks of USF1 and USF2 in H1 cells. Using chromatin status annotation (i.e., ChromHMM), we found that the overlapping peaks of both TFs showed different profiles compared to the overall ChIP-seq peaks, and were mostly enriched in weak enhancer regions (Fig. 7d). Furthermore, removal of ChIP-seq peaks with moderate to high methylation did not affect this observation. These results suggested that USF1

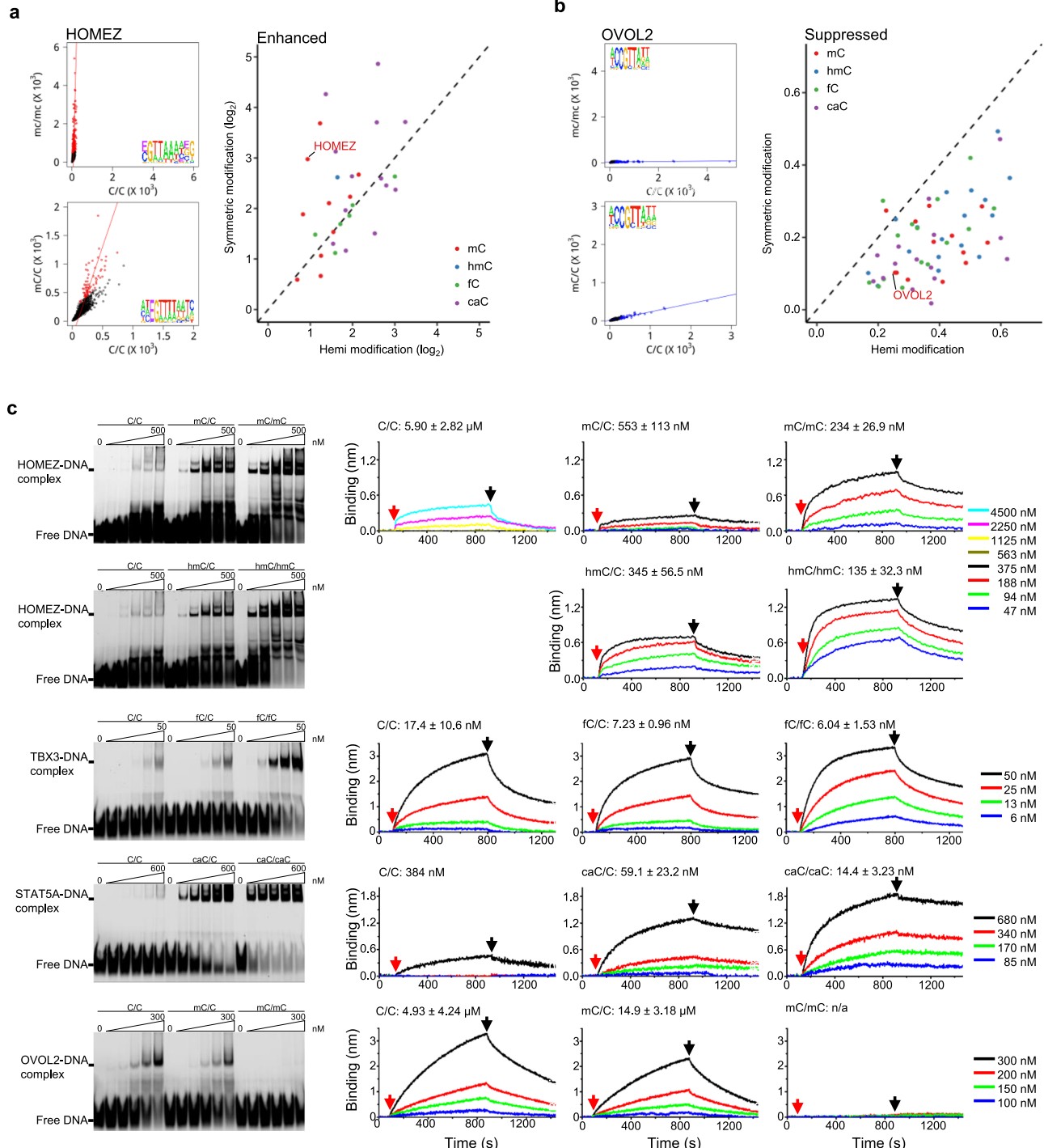

**Fig. 6 Symmetric modifications have a greater impact on TF-DNA interactions than hemi-modifications. a** Symmetric modifications enhance TF's binding to modified consensus sequences more than hemi-modifications. The enriched 6-mers (red dots) show a steeper slope for symmetric methylation than hemi-methylation, although methylation in both symmetric and hemi-forms enhanced HOMEZ's binding activity. In 16 (53.33%) of the 30 modification-enhanced cases symmetric modifications showed higher slopes than the hemi-modifications. Letter E in the consensus sequences represents modified cytosine. **b** Hemi-modifications suppress TF's binding to modified consensus sequences less than symmetric modifications. The suppressed 6-mers (blue dots) show a flatter slope for symmetric methylation than hemi-methylation for OVOL2. In 58 (96.7%) of the 60 modification-suppressed cases symmetric modifications showed flatter slopes than the hemi-modifications. **c** Binding kinetics and affinity studies. Electrophoretic mobility shift assays (EMSA) were used to obtain semi-quantitative measurements of interactions between modified DNA motifs and the corresponding TFs at various concentrations. The DNA probe for HOMEZ's EMSA validation was designed by assembling the six 6-mer sequences with the highest frequency. The OCTET system was employed to determine binding kinetics and affinity values for DNA-TF interactions as described above. The red and black arrows indicate when the DNA probe immobilized on an OCTET biosensor was dipped into TF solution and wash buffer, respectively. The deduced $K_D$ values are listed for each assay performed at various TF concentrations shown in different colors. DNA oligo sequences used for the EMSA and OCTET were shown in Supplementary Table 1. Source data are provided as a Source Data file.

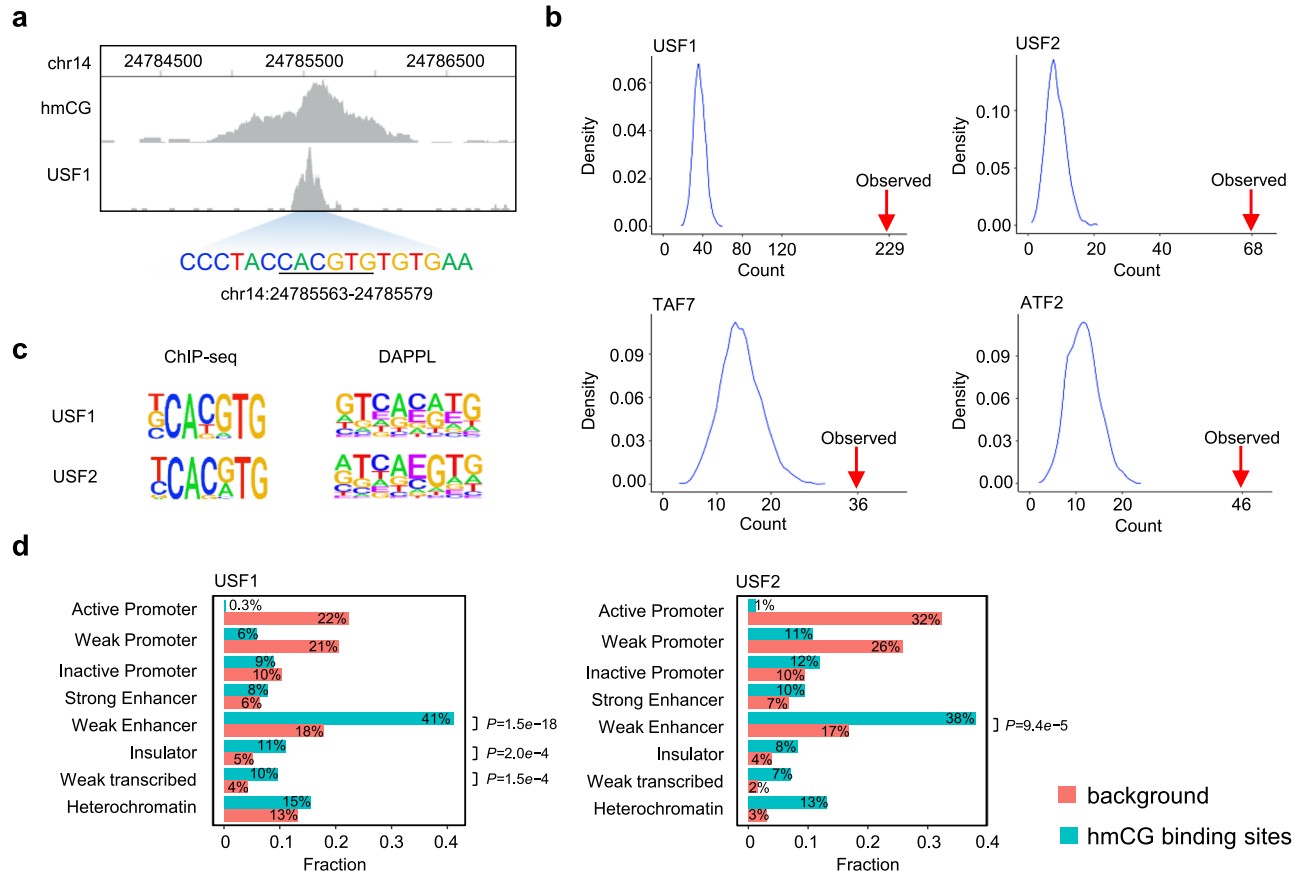

**Fig. 7 Potential roles of hmCG readers in human embryonic stem cells. a** A genomic region of overlapping hmC and USF1's ChIP-seq peaks. The identified USF1's hmC consensus found in the peak region is underlined. **b** Statistical significance of the overlapping peaks. 5000 random assignments of TF ChIP-seq peaks in the genome produced a distribution of the numbers of peaks overlapping with hmC peaks. The observed numbers of the overlapping peaks were indicated with the red arrows for USF1, USF2, TAF7, and ATF2. **c** Comparison of the consensus sequences obtained with DAPPL and those with sequences in the overlapping peaks. Letter E in the consensus sequences represents modified cytosine. **d** Distribution of chromatin states of the overlapping peaks (blue bars) compared with all of the ChIP-seq peaks (red bars). The overlapping peaks of both USF1 and USF2 were enriched in weak enhancer regions with significant P values. P values were calculated by two-sided Chi-square without adjustments. Source data are provided as a Source Data file.

and USF2 might regulate transcription via hmC-binding activity in weak enhancers in H1 cells.

## Discussion

In this study, we developed an all-to-all approach, DAPPL, to enable highly multiplexed profiling of protein-DNA interactions in the context of DNA epigenetic modifications. To benchmark this approach, we focused on the ETS subfamily because almost all of them have well-characterized consensus sequences. On the basis of our results and analyses, we demonstrated that DAPPL approach could reproducibly recover all known consensus sequences with good quality in a single round of selection. Of note, many TFs, such as the bZIP family members, are known to form homo- or heterodimers before they can bind DNA[49]. While the DAPPL approach cannot be used to identify consensus sequences for heterodimers, it might work for homodimers because some TFs are obligatory homodimers. For example, ETS1 in full-length is not known to bind DNA as a monomer[50,51]. However, the two ETS1 isoforms both identified specific DNA motifs in our pilot DAPPL assays (Supplementary Data 2). A likely explanation is that such TFs might have already formed homodimers before they were captured on the glutathione beads.

Conceptually, the establishment of this all-to-all approach should revolutionize traditional high-throughput screening, because most of the current high-throughput approaches come in the form of one-to-all. DAPPL represents a technology

breakthrough because its multiplicity (e.g., 192 proteins vs. $2.15 \times 10^{10}$ DNA species) far exceeds any existing multiplexed methods, such as dye-based approaches[52]. Our DAPPL technology allowed us to assay a mixture of five randomized DNA libraries carrying four different cytosine modifications and unmodified cytosine, and the unique advantage of such a competition assay allowed us to directly compare binding preferences of a given TF (e.g., Fig. 4 and Supplementary Fig. 7).

We believe that DAPPL could have a wide range of applications and can be adapted by the scientific community for various studies. For example, DAPPL can be tweaked to apply to the study of protein-RNA interactions. Given >100 different RNA modifications identified so far, we expect that the adaptation of DAPPL in studying RNA posttranscriptional modifications will have a long-lasting impact. Moreover, DAPPL can be applied to rapidly identify small molecule binders for drug targets using DNA-encoded small molecule libraries (i.e., DEL), some of which have reached the complexity of over a trillion compounds. We anticipate that simultaneous forward and counter screenings in a single test tube enabled by DAPPL will greatly speed up drug discovery, while reducing unwanted toxicity in order to improve the success rate during clinical trials.

To our satisfaction, our DAPPL approach recovered several known methylated cytosine readers, such as methylated DNA readers MeCP2, MBD4, MEIS1, HOMEZ, TBX2/3/20, RFX2, and PKNOX2[20,41,53]. In two recent studies, DNA baits, carrying mC,

hmC, fC, or caC modifications, were used to pull down potential binding proteins in cell lysates to identify potential modification readers[20,42]. Although the binding specificity of these readers was not known, quite a few of them were also recovered in our study, such as a hmC reader MeCP2, seven fC readers (CNBP, CSDA, GTF2I, NRF1, PURA, FOXP4, and p53), and seven caC readers (ZBTB7B, CTCF, NRF1, SMARCC1, ZBTB7A, ZBTB7B, and ZNF187)[20,23,42]. Except MeCP2, CTCF, and p53, we identified the consensus sequences carrying the corresponding modifications for the rest of the readers, demonstrating the advantage of the DAPPL approach.

We also compared our results with the report by Yin et al.[41]. First, 88 and 265 proteins were found to recognize methylated motifs in this study and Yin et al., respectively[41]. 43 were identified in both studies ($P = 0.0078$). These studies both uncovered that the Homeodomain TFs were found to prefer methylated DNA motifs. On the other hand, methylation was observed to significantly suppress binding activity of the ETS family in both studies. In Mann et al., methylation was reported to enhance CEBPα and CEBPβ's binding activities, and inhibit CREB, ATF4, JUN, JUND, CEBPδ, and CEBPγ. Of these eight proteins, CEBPβ, CEBPδ, CEBPγ, ATF4, and JUN were also assayed in this study. We identified significant motifs for CEBPβ and CEBPγ using the symmetric-methylated DNA library but did not find any significant enhancement or suppression by methylation. Although both proteins were categorized as "MethylPlus" in Yin et al., the effects were only marginal[41]. Therefore, our studies, in agreement with many previous studies, suggest that cytosine modification enhancement or suppression of TF-DNA interactions might be a widespread phenomenon in humans.

Because the TFs were sampled against the five libraries with different modifications simultaneously in the DAPPL assays, we were able to predict whether a modification in a specific sequence context would enhance or suppress a TF-DNA interaction. Using both EMSA and quantitative kinetics studies, all of the tested predictions were successfully verified (Fig. 6 and Supplementary Fig. 9). Interestingly, Yin et al. also employed bioinformatics analysis to predict those TF-DNA interactions that could be enhanced (i.e., MethylPlus) or suppressed by symmetric DNA methylation (i.e., MethylMinus), although no experimental validation was provided[41]. Of the 14 TFs identified as methylation-enhanced cases in our study, three proteins, PKNOX2, MEIS2, and MEIS3, were also identified as MethylPlus; whereas the other 11 TFs were not included in the study by Yin et al. One of these 11 proteins, HOMEZ was confirmed to bind to the hemi and symmetric-mC-carrying consensus sequences 10- and 25-fold stronger than the unmodified counterpart, respectively (Fig. 6c). Of the 19 methylation-suppressed TFs identified in this study, 11 were also included in the Yin et al. study. Eight of them were also scored as MethylMinus. OVOL2's binding activity to symmetric-methylated DNA was undetectable, while its binding affinity dropped to 15 μM to hemi-methylated DNA, 3-fold weaker as compared with the unmethylated counterpart (Fig. 6c).

Compared with intensive studies of symmetric modifications, the biological roles of hemi epigenetic modifications remain obscure and underexplored[25]. Emerging evidence suggests that hemi-mC is a stable epigenetic mark because a small fraction of hemi-mC could be inherited, and it has been associated with transcriptional regulation[24]. Moreover, in a non-CpG context (e.g., CpA, CpT, or CpC) mC is asymmetrical and mCpA, in particular, is often found in gene bodies and correlated with gene transcription in mammalian stem cells and neurons[31,33]. However, the molecular functions of such hemi epigenetic marks remain largely unknown because of a lack of readers. Previously, only a handful of proteins, such as Dnmt1 and UHRF1, were reported to recognize hemi-mC in cells[54,55]. We identified 18, 11,

15, and 24 proteins could preferentially recognize consensus sequences carrying hemi-mC, -hmC, -fC, and -caC, respectively (Supplementary Data 8 and Supplementary Fig. 7). Consistent with results observed with symmetric modifications, many hemi-modifications could either enhance or suppress TF-DNA interactions, although to a lesser extent as evidenced by kinetics studies. We believe that our results and analyses will pave the way for the elucidation of the physiological roles of these types of epigenetic marks.

## Methods

**Transcription factor clones and protein purification**. Using the existing human ORF expression library, each TF protein was expressed in yeast, purified as N-terminal GST fusion, and captured on glutathione beads[56]. A step-by-step protocol describing the protein purification with a high-throughput method can be found at Protocol Exchange[57]. In brief, each yeast strain was grown in 800 μL of SC-Ura/glucose liquid medium overnight as primary cultures. Fifty microliter of saturated yeast cultures were inoculated into 8 mL of SC-Ura/raffinose (Sigma–Aldrich) liquid medium until the O.D. 600 reached 0.6, followed by induction with 2% galactose (Sigma–Aldrich) for 6 h at 30 °C. The harvested yeast cell pellets were stored at −80 °C or immediately lysed in lysis buffer (50 mM Tris-HCl at pH 7.5 with 100 mM NaCl, 1 mM EGTA, 0.01% tritonX-100, 0.1% beta-mercaptoethanol, 1 mM PMSF, and Roche protease inhibitor tablet [Roche]). Next, the cell lysates were centrifuged at 3,500 g for 10 min at 4 °C; the supernatants were transferred to a plate with prewashed glutathione beads (GE Healthcare), and incubated overnight at 4 °C to capture the GST fusion proteins. Subsequently, the beads were washed three times with wash buffer I (50 mM HEPES at pH 7.5 with 500 mM NaCl, 1 mM EGTA, 10% glycerol, and 0.1% beta-mercaptoethanol) and three times with wash buffer II (50 mM HEPES at pH 7.5 with 100 mM NaCl, 1 mM EGTA, 10% glycerol, and 0.1% beta-mercaptoethanol). All TFs conjugated to glutathione beads were stored at −80 °C until use. As a negative control, GST tag was expressed and purified with the other transcription factors.

When necessary, the TF proteins were eluted from glutathione beads with elution buffer (50 mM HEPES at pH 8.0 with 100 mM NaCl, 50 mM KAc, 5 mM MgCl₂, 40 mM glutathione, 10% glycerol, and 0.05% Tween-20). For the TF-DNA-binding kinetics assay, glycerol (a high refractive index component) was left out of the elution buffer (50 mM HEPES at pH 8.0 with 100 mM NaCl, 50 mM KAc, 5 mM MgCl₂, 40 mM glutathione, and 0.05% Tween-20).

**Conjugation of anchor oligo to TF proteins in a 96-well format**. A step-by-step protocol describing the conjugation of DNA oligos to TF proteins can be found at Protocol Exchange[57]. To barcode TF proteins with DNA, a common oligo (i.e., the anchor oligo) was first covalently conjugated to each TF protein. Before the conjugation assay, the maleimide-2,5-dimethylfuran cycloadduct on the 5′-end of the anchor oligo (Genelink) was converted to maleimide via a Retro-Diels-Alder reaction. After cooling to room temperature, the lyophilized oligos were dissolved in PBS buffer to a final concentration of 50 μM and added to each purified protein on glutathione beads arrayed in 96-well plates. The conjugation was achieved using a "click" chemistry reaction between the hydro sulphonyl group on cysteine residues of the proteins and a maleimide group tethered to the 5′-end of the anchor oligo at room temperature for 1 h. The free oligos were removed with three stringent washes (50 mM HEPES at pH = 7.5 with 100 mM NaCl, 1 mM EGTA, 10% glycerol, and 0.1% beta-mercaptoethanol) and the anchor oligo-conjugated TFs were stored at −80 °C until use.

**Assignment of DNA barcodes to TF proteins**. A step-by-step protocol describing the assignment of DNA barcodes to TF proteins can be found at Protocol Exchange[57]. To assign a unique DNA barcode to each protein, a collection of 2000 address oligos (Integrated DNA Technologies) were synthesized, each of which containing a *Bsa*I recognition site and a cutting site (GGTCTCCGACT) at the 5′-end, a 8 nt random sequence as unique molecular identifiers (UMI), a 7–11 nt unique DNA barcode sequence, and a 20-nt consistent sequence, complementary to the anchor oligo (Supplementary Fig. 1a and Supplementary Table 1). Next, the address oligos were individually annealed to the anchor oligos conjugated on the TF proteins in a 96-well format, followed by a Klenow polymerase reaction (1 × NEB Buffer 2 with 0.6 mM dNTP mix, 1 U Klenow [New England Biolabs], 12 uM address oligos) at 37 °C for 30 min to synthesize the complementary strands. Free address oligos were removed with three washes in the same washing buffer described above. 1/50 of the bead slurry of each protein was taken from two adjacent plates (i.e., 192 = 2 × 96) and pooled. A total of eight protein pools were generated for the DAPPL reactions.

**DNA library preparation**. The 16-mer random region of template DNA oligos (Integrated DNA Technologies) was synthesized in an A:C:G:T ratio of 30:30:20:20 because this ratio is known to provide a more equal distribution of the four bases[58–60]. The 16 random nucleosides were designed to be flanked by two consistent sequences: one for amplifying DAPPL product (i.e., 5′-GGGAGAAGG

TCATCAAGAGG) and the other with a *Bsa*I restriction site and its cutting site (i.e., 5′-GGCATGCAGCCACTATAAGCTTCGAAGACTTGAGACCAT). The double-stranded 16-mer DNA library was generated by annealing the template oligo with a complementary primer oligo in a 1:1 ratio, followed by a Klenow polymerase reaction. By design, after *Bsa*I digestion the sticky ends of the 16-mer DNA library were complementary to those of the proteins' DNA barcodes such that they can be annealed and ligated when brought in close proximity.

To create DNA libraries carrying four different epigenetic modifications, including mCpG, hmCpG, fCpG, and caCpG, the template oligos (Genelink) were first synthesized to encode a 5′ Primer 2 (5′-CACATCCTTCACATTAATCC), an 18-mer sequence with a modified CpG in the middle flanked by two 8-mer random sequences, a short sequence encoding the library identity, and a *Bsa*I recognition site and its cutting site (Supplementary Table 1). These oligos were then converted into dsDNA libraries in the presence or absence of the modified dCTPs to create symmetric or hemi-libraries, respectively, using the Klenow reactions as described above. Each library was purified with a QIAquick PCR Purification kit (Qiagen) according to the manufacturer's instructions. The design of each modified library is shown in Supplementary Table 1. In addition, an unmodified DNA library of the same design was also created as a reference library.

Finally, the four libraries with four different symmetric modifications and the unmodified library were mixed in an equimolar ratio to form the mixture of symmetric modification libraries. A mixture of the hemi-modification libraries was created using the same method in parallel.

**Establishment of DAPPL method.** A step-by-step protocol describing the procedure of DAPPL assay can be found at Protocol Exchange[57]. To develop and optimize the digital affinity profiling via proximity ligation (DAPPL) reactions, we incubated each TF mixture on beads with 200 nmole of the 16-mer DNA library in a TF-binding buffer (10 mM Tris-HCl at pH 7.5 with 50 mM NaCl, 1 mM DTT, and 4% glycerol) at room temperature for 30 min. After three stringent washes in binding buffer and PBS buffer, the protein-DNA complexes on the beads were crosslinked by 0.1% formaldehyde for 10 min and quenched with Tris-Glycine buffer (pH 7.5). Next, the beads were washed in TBST buffer (0.01% tween-20) three times and equilibrated by 1 × T4 DNA ligase buffer. A Golden Gate Assembly reaction was performed by adding a master reaction mixture (227 μL of ddH$_2$O with 30 μL 10 × T4 DNA ligase reaction buffer, 30 μg bovine serum albumin, 20 μL of 100 U of *Bsa*I [New England Biolabs], and 20 μL of 600 U T4 DNA ligase [Enzymatics]) to the beads and incubated for 1 h at 37 °C[43]. After Proteinase K (New England Biolabs) treatment and phenol/chloroform (ThermoFisher Scientific) extraction, the ligated DAPPL products were subjected to 15–18 cycles PCR amplification (New England Biolabs) to prepare the sequencing libraries, followed by a gel extraction step. PCR cycles were determined on the basis of the final quantity of the PCR products, which was about 500 ng. A Next-Generation sequencing library was constructed and deep-sequenced on an Illumina NextSeq500 sequencer.

**DAPPL to discover readers for different epigenetic DNA modifications.** The optimized protocols described above were applied to identify potential readers for symmetric and hemi-modifications using 200 nmole of the mixture of symmetric or hemi-modification libraries, respectively. Four random DNA libraries were synthesized, each of which carried a CpG site with mC, hmC, fC, or caC flanked by two 8-mer random sequences. A three-nucleotide barcode was also added to each sequence to identify the modification (Supplementary Table 1). To ensure symmetric modifications at the middle CpG sites, the complementary strands of each library were synthesized with Klenow polymerase in the presence of the corresponding modified dCTPs (i.e., 5-methyl-dCTP; 5-hydroxymethyl-dCTP; 5-formyl-dCTP; 5-carboxy-dCTP). For comparison, an unmodified library of the same design was also synthesized (Supplementary Table 1). An equal amount of the five DNA libraries was pooled together to generate the symmetric modification DNA reaction mixture (Fig. 1). In parallel, the four hemi-modified DNA libraries and the unmodified library were also synthesized and mixed in an equimolar ratio. The two library mixtures carrying the symmetric or hemi-modifications were then separately incubated with the TF mixtures to carry out the DAPPL reactions, as described above. A light crosslinking step was also included before the proximity ligation, as described above. Since formaldehyde-based crosslinking is known to prefer primary amines on G/C/A nucleotides[61], it is unlikely for the Schiff base to react with the methyl, hydroxymethyl, formal, or carboxyl moieties on the modified cytosine. Note that we did not remove the modifications before deep-seq; the type of modification could be distinguished based on the built-in library barcodes (Supplementary Table 1).

**Generation of PWM models.** Because each DAPPL product carried the barcode sequence of the TF that captured this sequence (see Fig. 1), all of the TF-captured sequences were partitioned by the TFs (via the TF barcodes). Scripts were programmed in R language. The seqinr and Biostrings packages were called to read/write the fasta/fastq files and match TF barcodes. As this information was kept throughout the entire computational analysis, the sequencing reads could be readily mapped back to each TF. Since each TF was fused with GST, we first excluded the DNA-binding activity contributed by the GST tag. To do so, we first

extracted the 6-mers by a sliding window of length 6 moving along the sequencing reads one nucleotide at a time. The 6-mer occurrences were obtained for the binding sequences to a particular TF. Similarly, the 6-mer occurrences were also obtained for the binding sequences for the GST proteins, which were included in each multiplexed binding assay as a negative control. Next, we compared the 6-mer sequence occurrences between those obtained with each TF and the GST counterparts. We then determined the 6-mers that were enriched for a given TF by comparing their occurrences with their GST counterparts.

The count of the 6-mers bound by GST was referred as:

$$\mathbf{X} = \{x_1, x_2, \ldots \ldots x_{4096}\} \tag{1}$$

The count of the 6-mers bound by a TF was referred as:

$$\mathbf{Y} = \{y_1, y_2, \ldots \ldots y_{4096}\} \tag{2}$$

The occurrence pairs of the 6-mers were referred as:

$$\mathbf{P} = \{p_1(x_1, y_1), \ldots \ldots p_{4096}(x_{4096}, y_{4096})\} \tag{3}$$

We defined the enriched 6-mers for each protein using the following criteria. First, its slope (i.e., the ratio of the 6-mer frequencies of a given protein over GST) had to be greater than 1. Second, we selected the top 25th percentile of 6-mers with slopes >1. Finally, only the 6-mers whose frequencies were greater than the median was selected. Therefore, the enriched 6-mers were determined using the following equations.

$$\mathbf{M} = \{(x_i, y_i) | x_i < y_i\} \tag{4}$$

$$\mathbf{N} = \left\{ (x_i, y_i) | \frac{y_i}{x_i} > \mathrm{Q}_3 \left( \left\{ \frac{y_m}{x_m} | (x_m, y_m) \in \mathbf{M} \right\} \right) \right\} \tag{5}$$

$$p_i \in \{(x_i, y_i) | y_i > \mathrm{median}(\{y_n | (x_n, y_n) \in \mathbf{N}\})\} \tag{6}$$

Where $\mathrm{Q}_3 \left( \left\{ \frac{y_m}{x_m} \right\} \right)$ refers to the 3rd quantile of the slops and median($\{y_n\}$) represents the median value of 6-mer frequency.

The collection of the enriched 6-mers was used to construct the binding consensus sequences (i.e., motifs) using HOMER (http://biowhat.ucsd.edu/HOMER). The enriched 6-mers were first mapped back to the original library sequences and these sequences were used as input values for HOMER. Each motif was associated with a *P* value. To determine the cutoff of the *P* values in identifying significant motifs, we performed 400 nonreturn random sampling on the GST-bound sequences with a sample size of 10,000 in each library. Each of the 400 runs generated a top *P* value. These 400 *P* values were considered as a null distribution and the *P* value at the 95 percentile (i.e., 0.05 false discovery rate) was used as the cutoff to calculate the significant motifs. These modified motifs were built using customized R script calling the ggseqlogo package.

**Identification of modification-dependent TF-DNA interactions.** We identified sequences that were specific to either modified or unmodified sequences by comparing the sequence frequencies from the modified versus unmodified library for a given protein. To correct the possible library-specific bias, we first performed the Lowess data normalization[62] for GST from different modified libraries. Here, $\mathbf{X} = \{x_1, x_2, \ldots x_n\}$ and $\mathbf{Y} = \{y_1, y_2, \ldots y_n\}$ were used to denote the 6-mer frequencies obtained from unmodified versus modified DNA libraries for GST proteins. A log$_2$-based scatterplot of 6-mer intensity ($\mathbf{A} = \log_2(\mathbf{X} + \mathbf{Y})$) versus ratio ($\mathbf{M} = \log_2(\mathbf{X/Y})$) was plotted. A local weighted linear regression was used to calculate a regression curve from the corresponding scatterplot. This curve was then used to correct systematic deviations of 6-mer frequencies between two libraries for each TF.

We then compared the adjusted frequencies of 6-mer obtained from modified library and unmodified library. By plotting the 6-mer frequencies obtained with modified versus unmodified DNA libraries, we identified those sequences that were preferentially bound by TFs in either modified or unmodified format. We next estimated the expected deviation of frequencies of 6-mers for GST proteins from diagonal line ($y = x$), which represents the difference between the modified and unmodified sequences. We calculated the distance ($Dist_{GSTi}$) between a 6-mer bound by GST and diagonal line $y = x$. We set the distance cutoff as:

$$\mathrm{Dist}_{\mathbf{cutoff}} = \mathrm{avg}(\mathbf{Dist_{GST}}) + 6 \times \mathrm{sd}(\mathbf{Dist_{GST}}) \tag{7}$$

where avg() and sd() present average and standard deviation of the distance distribution.

Please note that the corresponding *P* value is $1 \times 10^{-5}$ at an S.D. of 6. If a 6-mer is bound by a TF located outside the distance cutoff in the scatterplot (i.e., $Dist_{TFi} > Dist_{cutoff}$), it was considered as a candidate carrying either enhanced or suppressed activity to the TF. After we identified the candidate 6-mers, we calculated the log$_2$-based slope for each TF. If the log2-based slope is greater than 0.5 or smaller than $-0.5$, those 6-mers were used to construct the modification-specific consensus sequences.

**Quantifying similarity between motifs.** Tomtom[63] was used to quantify the similarity between two motifs. Two motifs with a *P* value < 0.01 were considered similar. When multiple motifs were associated with a TF in CIS-BP, we used the

smallest $P$ value. Note that we did not treat the modified cytosine differently when assessing motif similarity using Tomtom.

**Quantification of the TF-binding preference**. We used the slope of the enriched 6-mers in the scatterplot as a proxy to evaluate binding preference to either modified or unmodified sequences. The count of N enriched 6-mers in unmodified library was referred to as:

$$\mathbf{X} = \{x_1, x_2, \ldots \ldots x_N\} \qquad (8)$$

The count of the enriched 6-mers in modified library was referred to as:

$$\mathbf{Y} = \{y_1, y_2, \ldots \ldots y_N\} \qquad (9)$$

The slope was calculated as:

$$S = \frac{\sum_{i=1}^{N} \frac{y_i}{x_i}}{N} \qquad (10)$$

**Electrophoretic mobility shift assay**. The oligos of the DNA probes used for EMSA assays were designed and synthesized by IDT (Integrated DNA Technologies) and Genelink (Genelink) as shown in Supplementary Table 1. They were converted to dsDNA with a T7 primer end-labeled with Cy5 using the Klenow polymerase reaction in the presence or absence of the desired modified dCTPs. The resulting dsDNA were purified with QIAquick PCR Purification Kit.

To perform EMSA assays, a Cy5-labeled DNA probe at 5 nM was incubated with its corresponding TF protein(s) in a binding buffer (25 mM HEPES at pH 8.0 with 100 mM NaCl, 5% (v/v) glycerol, 50 mM KAc, 5 mM MgCl₂, 1 mM DTT and 0.1 mg/mL bovine serum albumin) at room temperature (~22 °C) for one hour. The reaction mixture was then loaded onto a 5% Criterion TBE Gel (BioRad), and electrophoresed for 90 min at 80 V in 0.5 X TBE buffer. Formation of protein-DNA complexes were detected by scanning the TBE gel using Odyssey® CLx Imaging System (LI-COR Image Studio Lite Ver 5.2, LI-COR Biosciences).

**TF-DNA-binding kinetics and affinity measurement with the OCTET**. The TF-DNA-binding kinetic assays were performed using OCTET RED96 device, equipped with SAX (High Precision Streptavidin Biosensor) biosensor tips (FortéBio). Biotin-labeled dsDNA probes were generated using the same method as described above but with a biotin-labeled T7 primer. Each DNA probe was then diluted to a final concentration of 0.1 μM in the DNA-binding buffer (50 mM HEPES at pH 8.0 with 100 mM NaCl, 50 mM KAc, 5 mM MgCl₂, 40 mM glutathione, 5 mg/mL BSA, and 0.05% Tween-20). Each TF protein of interest was purified from large yeast cultures and serial two-fold dilutions were made in the DNA-binding buffer. The binding kinetics of TF-DNA interactions was measured according to the manufacturer's instructions (FortéBio). In short, a streptavidin-coated biosensor was first immersed in the DNA-binding buffer for 10 min to establish the baseline, followed by dipping in the DNA probe well to capture the biotinylated DNA probe to the biosensor. After another approximate 600-sec baseline establishment in the binding buffer, the biosensor was dipped into a TF protein well to obtain the on-curve until signal saturation. The off-curve was obtained by transferring the biosensor to a well containing fresh binding buffer until the off-curve became flat. Data collection, data analysis and curve-fitting were performed using FortéBio's Data Acquisition 7.1 and Data Analysis 7.1, based on which the $K_{on}$ and $K_{off}$ values were determine for each binding assay. The $K_D$ values were deduced by taking the ratios of $K_{off}/K_{on}$.

Probes used for the OCTET experiments in Fig. 6c and Supplementary Fig. 9b are listed in Supplementary Table 1.

**Genome-wide hmC profiling of human embryonic stem cell H1**. Human embryonic stem cell H1 was purchased from WiCell Research Institute (WiCell) and the ethics approval was obtained from the Robert-Koch Institute, Berlin, Germany. Genomic DNA was isolated from human embryonic stem cell H1 (WiCell) with standard protocols. The cell pellet of two million H1 cells was suspended in 500 μL digestion buffer (100 mM Tris-HCl at pH 8.5 with 5 mM EDTA, 1% SDS, 200 mM NaCl) on ice and then treated with Proteinase K at 55 °C overnight. After phenol/chloroform (25:24:1 saturated with 10 mM Tris, pH 8.0, and 1 mM EDTA) extraction the aqueous phase was transferred to a test tube, mixed with same volume of isopropanol, and stored at −80 °C overnight to precipitate the genomic DNA. After spinning at 21,000 × g for 10 min at 4 °C, the DNA pellet was washed with 75% ethanol, air-dried and dissolved in Nuclease-free Water. To perform the hmC-seq, hmC-specific chemical labeling and enrichment of DNA fragments with hmC were performed using a previously described method[47]. DNA libraries were prepared following the Illumina protocol of 'Preparing Samples for ChIP Sequencing of DNA' (Illumina) using genomic DNA or hmC-captured DNA and then subjected to deep-seq on the Illumina Hi-seq 2000 machine.

**Mapping overlapping regions between hmC and TF ChIP-seq peaks**. For hmC-seq data, we aligned these fastq files into reference genome and used MACS2 to do peak calling separately for two repetitions (settings: macs2 callpeak -t input_file -f

BAM -g hs -n output_prefix -B -q 0.01). We used IDR (Irreproducible Discovery Rate) framework to measure consistency between two replicates. Those shared regions with IDR less than 0.05 we regard as consistent and reproducible peaks. In total, we identified 3892 hmCG modified regions. We downloaded all narrowPeak files about TFs ChIP-Seq datasets in H1-hESC cell lines from ENCODE and UCSC, which revealed seven TFs that have binding sites that overlap with hmCG modified regions. The numbers of overlapping regions were compared with those obtained from 5000 simulation of randomly selected genome regions of the same width distribution as the ChIP-seq peaks. USF1, USF2 and TAF7, and ATF2 were found to have enriched overlapping regions. Furthermore, significant motifs were identified for USF1 and USF2 using these overlapping regions (using MEME to call motifs).

Next, we investigated the chromatin status of TF binding under hmCG modification. After combining our data with the WGBS dataset, we regard the TF-binding regions with average methylation less than 0.2 and not overlapping with hmCG modified regions as background. Using ChromHMM annotation of H1-hESC, we found that TF binding under hmCG modification had different chromatin states (R package: GenomicRanges was used to do the overlapping analysis). The ChromHMM annotation was download from UCSC Genome Browser (http://hgdownload.soe.ucsc.edu/goldenPath/hg19/encodeDCC/wgEncodeAwgSegmentation/wgEncodeAwgSegmentationChromhmmH1hesc.bed.gz). Then, liftOver was used to convert the annotation file from hg19 to GRCh38.

**Reporting summary**. Further information on research design is available in the Nature Research Reporting Summary linked to this article.

## Data availability
All the raw and processed Sequence data that support the findings of this study have been deposited in the NCBI Gene Expression Omnibus (GEO) database under the accession codes [GSE160457]. Source data are provided with this paper.

## Code availability
All of the computer programs and scripts used are publicly available at https://github.com/HitTracy/DAPPL. A released vision of the code would be referred using the DOI (https://doi.org/10.5281/zenodo.4308235).

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

## Acknowledgements

This work was supported in part by the NIH (RO1 AG061852-03, RO1MH122451-01, RO1HL149961-01, R24 AA025017, U01CA200469, P50AA026116 to H.Z.; R01GM111514; to H.Z. and J.Q.; R01EY029548; R01EY024580 to J.Q.; R35NS116843 to H.S.; P01NS097206 to H.S. and P.J.; NS051630 and NS111602 to P.J. NIH Chemical Biology Interface Training Grant T32GM080189). We would like to thank Dr. Jef D. Boeke for his support and insightful discussion during this project and Dr. Jun Shen for providing us human embryonic stem cells. Finally, we would like to thank Jessica Dunn and Eric Johansen for proofreading this manuscript.

## Author contributions

G.S. conducted the majority of wet-lab experiments with help from Y.C., Q.S., C.M., X.L., and G.W. made the major contributions to developing bioinformatics algorithms and data analyses with help from J.W., J.B., H.S., and P.J. provided critical reagents and supports to this study. H.Z. and J.Q. conceived the idea, supervised this study and wrote the manuscript with the help from all the authors.

## Competing interests

The authors declare no competing interests.
