## [Peer Review File · Nature Communications]

REVIEWER COMMENTS

Reviewer #1 (Remarks to the Author):

Song et al developed an unbiased, highly multiplexed proximity-ligation based method, DAPPL, to study and profile TF-DNA interactions on epigenetically modified CpG dinucleotides, including methylated cytosine and its oxidized forms. Specifically, the authors surveyed human transcription factors (TF) with random DNA libraries carrying mC, hmC, fC, or caC modifications in either symmetric- or hemi-form to identify novel sequence- and modification-specific and effectors. The study revealed the differential impact of these four modifications on TF-binding capacity and specificity, and provided extensive amount of data to provide support for their DAPPL method. Although both the method and the data produced are highly valuable, some clarification and additional experiments will further strengthen and clarify their findings.

SPECIFIC COMMENTS:

1. Given the potential of PCR bias, how many cycles of PCR was performed to prepare the ligated DPPL products for sequencing?
2. How diverse was the starting random N-mer dsDNA library?
3. Many TFs form homo- or heterodimeric units to permit DNA binding or specify the target DNA domain. In some cases, monomeric TFs are unable to or rarely bind to DNA. For example, full-length ETS1 is not known to bind DNA as a monomer. Instead, ETS1 binds DNA only as a homo- or heterodimer with different partner proteins such as RUNX1, Pax-5, Pit-1, and NF- κ B. Thus, how do the authors mitigate such bias introduced by monomeric assembly of GST-tagged TFs on GSH beads? Did such caveats of dimeric TF-DNA binding inform selection of testable TFs?
4. Did the positive control HSF1-bound beads include assembly of the trimeric form?
5. What did the rest of the 50% sequencing reads yield (related to Figure 2a)?
6. Label the n for enriched 6-mers (red dots) for each TF. Perhaps, a heatmap density plot would be more informative. Same comment for other similar figures.

7. Past SELEX experiments have shown an increased enrichment for high affinity binders. However, it is intriguing that a second round of DAPPL did not significantly improve the quality of the consensus sequences. Wouldn't the authors expect that high affinity binding sequences would be enrichment with multiple cycles of DAPPL? Relatedly, a better presentation of Figure 2e would be useful.

8. To evaluate the precision of DAPPL, it would help to include TFs with mutated DNA binding domains.

9. After formation of the protein-DNA complex, three stringent washes are applied. Can the authors analyze the "washed/unbound" sequences after the second and third wash to identify potential transient or low affinity binding sequences?

10. Can the authors calculate and/or speculate the false positive and negative rate of DAPPL?

Reviewer #2 (Remarks to the Author):

This manuscript describes the development of a method to detect and quantify binding behavior of all human TFs against all 5mC oxidized forms of 5mC. This so-called "all-to-all" approach is high-throughput, and exploits random modified oligo library generation, unique tagging of proteins, and high-throughput sequencing. The method, called DAPPL, was then applied to all human TFs (or a good approximation of "all") and against random oligos with particular modifications (not necessarily "all", but also reasonable to me).

My assessment of the question is that it is inherently interesting

given recent results (still a bit controversial) about the functional relevance of hmC, fmC and caC (hmC seems now well established as correlate for certain aspects of gene regulation). I also believe the DAPPL approach is sound and has seen adequate controls: although one could imagine additional controls overall, I think the basic burden has been satisfied through the experiments done on the ETS family.

In terms of results, I would use these results as a catalogue or data resource, and in many individual cases (i.e. specific TFs vs. specific modifications) I suspect more focused studies would be required for confirmation. This does not diminish my view of the value of this as a data resource. These resources are very helpful when trying to interpret genome-wide epigenomic profiles and their (potential) relations to TF behavior. And as explained in the discussion, the general approach can be applied more broadly.

Major issues

-- The authors base much of their analysis about "stronger" or "weaker" trends on slope (e.g. Figs 5 & 6). In the context of hemi- vs. symmetric, is it possible to determine if there is an additive contribution to relative affinity?

-- I may have missed this (apologies if so), but in order for me to use these results in my own analyses, I would need a complete version of something like Table S4, in some format I can parse with a script, and that has quantitative information for all the possible entries in Table S4.

The availability of sequencing data in SRA allows that, in theory, for purpose of reproducibility, but why not just provide it so readers can actually use it directly. Also, with all possible motifs (and quantitative info) so in a given case a user can decide for themselves if the differences matter. I might be confused on why this was not done, as it seems so obvious.

Minor issues

-- What are the differences in types of information that can be obtained as a function of unique reads (random oligos) per protein? On orders of magnitude, I can imagine that 10k vs 100k vs 1M unique reads would allow dramatically different conclusions, even if interesting conclusions are attained at only smaller numbers.

-- The authors attempted multiple cycles for a reason -- but then noticed no improvements and moved on without comment. Was there an expectation of improvement in quality of results from multiple cycles? If so, do the authors have a sense of why results did not change?

-- The observation was made that no significant preference for a particular symmetric or hemi modification type was observed within each major TF subfamily, and that this suggests rapid evolution of this type of binding specificity. I find the claim confusing: what does this observation have to do with evolution?

-- I do not recall any comment about what is expected in terms of the fraction of reads that should contain the expected DAPPL products. Was

the reported 50% expected?

Reviewer #1 (Remarks to the Author):

Song et al developed an unbiased, highly multiplexed proximity-ligation based method, DAPPL, to study and profile TF-DNA interactions on epigenetically modified CpG dinucleotides, including methylated cytosine and its oxidized forms. Specifically, the authors surveyed human transcription factors (TF) with random DNA libraries carrying mC, hmC, fC, or caC modifications in either symmetric- or hemi-form to identify novel sequence- and modification-specific and effectors. The study revealed the differential impact of these four modifications on TF-binding capacity and specificity, and provided extensive amounts of data to provide support for their DAPPL method. Although both the method and the data produced are highly valuable, some clarification and additional experiments will further strengthen and clarify their findings.

SPECIFIC COMMENTS:

1. Given the potential of PCR bias, how many cycles of PCR was performed to prepare the ligated DPPL products for sequencing?

Response:

A fraction of each eluted DAPPL product was subjected to 15-18 cycles PCR amplification to prepare the sequencing libraries, followed by a gel extraction step. PCR cycles were determined on the basis of the final quantity of the PCR products, which was about 500 ng. In addition, UMI sequences were included in each TF barcode DNA to correct bias introduced by PCR reactions before the data analysis. We have added more detailed description to the Methods section (Second paragraph on page 15 and first paragraph on page 16).

2. How diverse was the starting random N-mer dsDNA library?

Response:

In theory, the expected diversity of the 16-mer dsDNA library should be 4.29×10^9 . However, we did not assess the diversity of the input 16-mer library using NextGen sequencing because it would require to obtain at least 13 billion reads to cover ~90% 16-mer species.

3. Many TFs form homo- or heterodimeric units to permit DNA binding or specify the target DNA domain. In some cases, monomeric TFs are unable to or rarely bind to DNA. For example, full-length ETS1 is not known to bind DNA as a monomer. Instead, ETS1 binds DNA only as a homo- or heterodimer with different partner proteins such as RUNX1, Pax-5, Pit-1, and NF- κ B. Thus, how do the authors mitigate such bias introduced by monomeric assembly of GST-tagged TFs on GSH beads? Did such caveats of dimeric TF-DNA binding inform selection of testable TFs?

Response:

The reviewer raised a very good point with regard to homo- and heterodimer formation among TF proteins. Indeed, ETS1 autoinhibition is mediated by the packing of helices within the N-terminal and C-terminal inhibitory domains onto helix H1 of the ETS domain (e.g., PMID: 26195629). In fact, the two ETS1 isoforms both identified very similar consensus sequences in our DAPPL reactions. A likely explanation is that these TFs already formed homodimers before they were captured on the glutathione beads. Another possibility, although unlikely, is that the GST moiety might affect the autoinhibitory domains, resulting in the release of the DNA-binding domain of ETS1. We are fully aware that our current design cannot determine the consensus sequences for TF heterodimers. Because we would like to screen as many TFs as possible in finding new epigenetic modification readers, we did not exclude those TFs that are known as obligatory dimers. We have added this point to the Discussion section (First paragraph on page 12).

4. Did the positive control HSF1-bound beads include assembly of the trimeric form?

Response:

Since we did not heat-shock the yeast strain during induction of HSF1 protein, it was unlikely the purified HSF1 protein was in the trimeric form (PMID: 26754925, PMID: 25963659, PMID: 16278218, PMID: 9727490). The fact that the identified consensus sequences of HSF1 were TTCnnGAA in both the pilot study (i.e., Supplemental Data 2) and the later screenings for epigenetic modification readers (i.e., Supplemental Data 4) suggested that HSF1 in our study was likely in the form of homodimer.

5. What did the rest of the 50% sequencing reads yield (related to Figure 2a)?

Response:

To answer this question, we re-analyzed the sequencing reads that were previously excluded in our analysis. A qualified DAPPL product had to precisely contain all the elements shown in Supplementary Figure 2a (also shown below). Any errors in the TF barcode sequences, Primer 1, Primer 2, ligation site, library ID, or the length of UMI would disqualify a DAPPL product (Supplementary Figure 2b). Note that the percentages in the table are not additive, because some DAPPL products removed from further analyses contained more than one error. We have now added this information to the main text (Last paragraph on page 5).

Supplementary Figure 2. Processing statistics of raw sequencing data.

a Expected structure of DAPPL products. **b** Percentages of the DAPPL products that have the expected elements. Note that the percentages in the table are not additive, because some DAAPL products removed from further analyses contained more than one error.

6. Label the n for enriched 6-mers (red dots) for each TF. Perhaps, a heatmap density plot would be more informative. Same comment for other similar figures.

Response:

It would be too busy to label the n for the enriched 6-mers (i.e., red dots) for each TF in Figures 2, 3 and 5 and therefore, we are providing all the data in Excel spreadsheets as part of the Data source. As suggested by the reviewer, we have generated the heatmap density plots for the ETS pilot study, as well as those of the following screenings for cytosine modification readers (Supplementary Figure 4 and Supplementary Data 6).

Supplementary Figure 4. Clustered heatmap density plots of 6-mers obtained with each TF.

a-b Heatmaps of 6-mers obtained with the 28 ETS, two positive controls (CRX and HSF1), and two negative controls (COPE and BCAT1) in cycle 1 (**a**) and cycle 2 (**b**) screenings. The scale bars represent the 6-mer frequency associated with a particular protein normalized by those obtained with GST. **c-d** Enlarged portions of the heatmaps where significantly enriched 6-mers were identified.

7. Past SELEX experiments have shown an increased enrichment for high affinity binders. However, it is intriguing that a second round of DAPPL did not significantly improve the quality of the consensus sequences. Wouldn't the authors expect that high affinity binding sequences would be enriched with multiple cycles of DAPPL? Relatedly, a better presentation of Figure 2e would be useful.

Response:

Thank you for providing us this opportunity to clarify. We have now performed additional analyses to compare the frequency of the enriched 6-mers (i.e. red dots in Fig. 2d) between cycle 1 and cycle 2 and found that these 6-mers were indeed further enriched in cycle 2 for every ETS protein, as well as the two positive control TFs (see Fig. 2f below). The fact that the similarity to CIS-BP consensus sequences of those consensus sequences obtained in cycle 2 did not show obvious improvement suggested that the DAPPL approach was sensitive enough to generate significant and reliable consensus sequences in one cycle of screening. One plausible explanation is that the ligation step in the DAPPL served as another layer of selection, resulting in further enrichment of the high affinity binding sequences. We have included this point to the main text, revised Figure 2f-h for a better presentation, and added Supplementary Figures 4 and 6 in the revised version.

Fig. 2f Comparison of the enriched 6-mers in cycles 2 and 1. The color represents the average frequency of the enriched 6-mers associated with a particular protein normalized by those obtained with GST.

Supplementary Figure 6. Comparison of 6-mer frequency obtained between cycle 1 and cycle 2 of the DAPPL reactions.

Heatmap plots of the 6-mer frequencies were generated for the 28 ETS, HSF1 and CRX in cycle 1 (left column) and cycle 2 (right column) of the DAPPL reactions. Each row

represents the 6-mer frequency associated with a particular protein normalized by those obtained with GST.

8. To evaluate the precision of DAPPL, it would help to include TFs with mutated DNA binding domains.

Response:

We agree with the reviewer that including TFs with mutated DNA binding domains would help evaluate the precision of DAPPL. Indeed, for this very reason we included CRX and HSF1 as two positive controls and two ubiquitin E3 ligases as negative controls in the DAPPL assays with the 28 ETS proteins. The fact that CRX and HSF1 both readily recovered their known consensus sequences that are very different from the ETS consensus sequence provided indirect evidence for the precision of DAPPL. In addition, the two E3 ligases failed to produce any significant consensus, indicating low background noises in the DAPPL reactions. Since use of mutant TFs is expected to produce negative results, it would be harder for us to interpret the negative results than the positive ones.

9. After formation of the protein-DNA complex, three stringent washes are applied. Can the authors analyze the “washed/unbound” sequences after the second and third wash to identify potential transient or low affinity binding sequences?

Response:

This would be a great idea for SELEX assays but would not work for DAPPL because a pool of the TF proteins were screened together in the same test tube. Because the unbound DNA fragments in the second and third washes were not ligated to the TF barcodes, we would not know from which TF protein(s) they were washed off. Although parallel DAPPL reactions could be set up to test this idea, it is beyond the scope of this study.

10. Can the authors calculate and/or speculate the false positive and negative rate of DAPPL?

Response:

This is a good question. The false positive rate of DAPPL can be estimated by the success rate of EMSA validation. Of the 50 EMSA assays we performed, 47 confirmed the DAPPL results, suggesting a false positive rate of 6%. The false negative rate, if estimated with the results of the ETS pilot study, would be 0% because all of the tested proteins could recover their known logos. A more rigorous estimate is to use the results of the unmodified CG library in the DAPPL screenings for epigenetic readers. Since 81

of the 751 known logos were recovered, the false negative rate is 89%. However, this is an overestimate because it was performed in a competition assay with four other modified libraries.

Reviewer #2 (Remarks to the Author):

This manuscript describes the development of a method to detect and quantify binding behavior of all human TFs against all 5mC oxidized forms of 5mC. This so-called "all-to-all" approach is high-throughput, and exploits random modified oligo library generation, unique tagging of proteins, and high-throughput sequencing. The method, called DAPPL, was then applied to all human TFs (or a good approximation of "all") and against random oligos with particular modifications (not necessarily "all", but also reasonable to me).

My assessment of the question is that it is inherently interesting given recent results (still a bit controversial) about the functional relevance of hmC, fmC and caC (hmC seems now well established as correlate for certain aspects of gene regulation). I also believe the DAPPL approach is sound and has seen adequate controls: although one could imagine additional controls overall, I think the basic burden has been satisfied through the experiments done on the ETS family.

In terms of results, I would use these results as a catalogue or data resource, and in many individual cases (i.e. specific TFs vs. specific modifications) I suspect more focused studies would be required for confirmation. This does not diminish my view of the value of this as a data resource. These resources are very helpful when trying to interpret genome-wide epigenomic profiles and their (potential) relations to TF behavior. And as explained in the discussion, the general approach can be applied more broadly.

Major issues

1. The authors base much of their analysis about "stronger" or "weaker" trends on slope (e.g. Figs 5 & 6). In the context of hemi- vs. symmetric, is it possible to determine if there is an additive contribution to relative affinity?

Response:

Our EMSA and binding kinetic studies shown in Figure 6 are in general agreement with the "slope" analysis. For example, the measured affinity value of HOMEZ to mC/C is about half of that to mC/mC (553 vs. 234 nM). However, TBX3's affinity to fC/fC is only 18% lower than that to fC/C. We believe that more data points will have to be collected to draw a definitive conclusion.

2. I may have missed this (apologies if so), but in order for me to use these results in my own analyses, I would need a complete version of something like Table S4, in some format I can parse with a script, and that has quantitative information for all the possible entries in Table S4. The availability of sequencing data in SRA allows that, in theory, for purpose of reproducibility, but why not just provide it so readers can actually use it directly. Also, with all possible motifs (and quantitative info) so in a given case a user can decide for themselves if the differences matter. I might be confused on why this was not done, as it seems so obvious.

Response:

We thank the reviewer for the suggestion. We provided position probability matrix of every TF's binding in Supplementary Data 3 and Supplementary Data 5. We uploaded all the raw and processed data to NCBI GEO database (GEO: GSE160457).

Minor issues

3. What are the differences in types of information that can be obtained as a function of unique reads (random oligos) per protein? On orders of magnitude, I can imagine that 10k vs 100k vs 1M unique reads would allow dramatically different conclusions, even if interesting conclusions are attained at only smaller numbers.

Response:

This is an intriguing and complicated question. In the ETS pilot experiment, we observed that the qualified reads associated with each protein varied dramatically, ranging from 16,317 to 313,382 (Fig. 2a). The fact that all of these proteins could recover their known consensus sequences suggested that the number of unique reads does not significantly affect finding the consensus sequences. For example, although the sum of UMI reads obtained for CEBPB and CRX from the symmetric mCG library were as low as 954 and 7,609, respectively, both produced robust consensus sequences. On the other hand, HMGN1 and LAS1L failed to generate any significant consensus sequences, even though their UMI reads were as high as 18,856 and 16,695, respectively.

4. The authors attempted multiple cycles for a reason -- but then noticed no improvements and moved on without comment. Was there an expectation of improvement in quality of results from multiple cycles? If so, do the authors have a sense of why results did not change?

Response:

We are sorry about the lack of clarification in the manuscript. To fully address your question, we performed additional analysis to compare the frequency of the enriched 6-mers (i.e. red dots in Fig. 2d) between cycle 1 and cycle 2 and found that these 6-mers were indeed further enriched in cycle 2 for every ETS protein, as well as the two positive controls (see Fig. 2f below). The fact that the similarity to CIS-BP consensus sequences of those consensus sequences obtained in cycle 2 did not show obvious improvement suggested that the DAPPL approach was sensitive enough to generate significant and reliable consensus sequences in one cycle of screening. One plausible explanation is that the ligation step in DAPPL served as another layer of selection, resulting in further enrichment of the high affinity binding sequences. We have included this point to the main text, revised Figure 2f for a better presentation, and added Supplemental Figures 4 and 6 in the revised version.

Fig. 2f Comparison of the enriched 6-mers in cycles 2 and 1. The color represents the average frequency of the enriched 6-mers associated with a particular protein normalized by those obtained with GST.

Supplementary Figure 6. Comparison of 6-mer frequency obtained between cycle 1 and cycle 2 of the DAPPL reactions.

Heatmap density plots of the 6-mer frequencies were generated for the 28 ETS, HSF1 and CRX in cycle 1 (left column) and cycle 2 (right column) of the DAPPL reactions.

Each row represents the 6-mer frequency associated with a particular protein normalized by those obtained with GST.

5. The observation was made that no significant preference for a particular symmetric or hemi modification type was observed within each major TF subfamily, and that this suggests rapid evolution of this type of binding specificity. I find the claim confusing: what does this observation have to do with evolution?

Response:

To avoid confusion, we have removed the half sentence about “rapid evolution.” (Last sentence of the second last paragraph on page 7)

6. I do not recall any comment about what is expected in terms of the fraction of reads that should contain the expected DAPPL products. Was the reported 50% expected?

Response:

To answer this question, we re-analyzed the sequencing reads that were excluded in our analysis. A qualified DAPPL product had to precisely contain all the elements shown in Supplementary Figure 2a as shown below. Any errors in TF barcode sequences, Primer 1, Primer 2, ligation site, library ID, or the length of UMI would disqualify a DAPPL product (Supplementary Figure 2b). Note that the percentages in the table are not additive, because some sequencing reads could be removed due to multiple sources. We have now added this information to the main text (last paragraph on page 5).

Supplementary Figure 2. Processing statistics of raw sequencing data

a The expected structure of DAPPL products. **b** Percentages of the DAPPL products that have the expected elements. Note that the percentages in the table are not additive, because some DAAPL products removed from further analyses contained more than one error.

REVIEWERS' COMMENTS

Reviewer #1 (Remarks to the Author):

The authors have addressed all my concerns. I have no more comments.

Reviewer #2 (Remarks to the Author):

All of my comments have been addressed.

Reviewer #1 (Remarks to the Author):

The authors have addressed all my concerns. I have no more comments.

Response:

We would like to thank the reviewer for the insightful comments and suggestions to improve this manuscript.

Reviewer #2 (Remarks to the Author):

All of my comments have been addressed.

Response:

We would like to thank the reviewer for the insightful comments and suggestions to improve this manuscript.